# Identification and characterization of small molecule inhibitors of the LINE-1 retrotransposon endonuclease

Alexandra M. D'Ordine [1,2], Gerwald Jogl [1,2] ✉ & John M. Sedivy [1,2] ✉

The long interspersed nuclear element-1 (LINE-1 or L1) retrotransposon is the only active autonomously replicating retrotransposon in the human genome. L1 harms the cell by inserting new copies, generating DNA damage, and triggering inflammation. Therefore, L1 inhibition could be used to treat many diseases associated with these processes. Previous research has focused on inhibition of the L1 reverse transcriptase due to the prevalence of well-characterized inhibitors of related viral enzymes. Here we present the L1 endonuclease as another target for reducing L1 activity. We characterize structurally diverse small molecule endonuclease inhibitors using computational, biochemical, and biophysical methods. We also show that these inhibitors reduce L1 retrotransposition, L1-induced DNA damage, and inflammation reinforced by L1 in senescent cells. These inhibitors could be used for further pharmacological development and as tools to better understand the life cycle of this element and its impact on disease processes.

The long interspersed nuclear element-1 (LINE-1 or L1) retrotransposon comprises ~17% of the human genome[1]. The L1 "mobile DNA" element propagates using an RNA intermediate in a copy-and-paste mechanism known as retrotransposition. L1 is the only autonomously replicating retrotransposon in humans, as it encodes the proteins required for its retrotransposition[2]. While most L1 sequences in the human genome are truncated or mutated, a small number (80-100) retain the ability to create additional L1 insertions[3] and mobilize non-autonomous retrotransposons, including Alu elements, which make up another 11% of the human genome[1]. New L1 or L1-driven germline insertions occur in up to one in every 20 individuals[4] and can cause disease[5]. Somatic L1 activity can also be harmful, as L1 has been implicated in cancer, neurodegeneration, and other age-associated diseases[6,7]. For example, L1 can create mutations, deletions, and other rearrangements when inserting a new copy[8,9], as well as acute DNA damage and cytotoxicity[10–14]. More recently, L1 expression has been associated with inflammation in senescent cells[15,16]. These cells have permanently exited the cell cycle and accumulate in aged tissues, where they contribute to inflammatory processes through the senescence-associated secretory phenotype (SASP)[17]. As these cells continue to persist, L1 is

derepressed and reinforces this inflammatory phenotype by triggering a type-I interferon (IFN-I) response[16,18]. Therefore, L1 can damage human health due to its ability to generate genomic instability and inflammation.

L1 encodes three proteins from three open reading frames: ORF0, ORF1, and ORF2. ORF0 is a 7 kDa primate-specific protein of unknown function located on the antisense strand[19]. ORF1 is a 40 kDa trimeric RNA-binding protein that binds the L1 transcript and is required for L1 retrotransposition[20,21]. ORF2 is a 150 kDa multi-functional protein that contains the enzymatic activities needed for retrotransposition: an N-terminal apurinic/apyrimidinic (AP)-like endonuclease (EN) domain[22], a reverse transcriptase (RT) domain[23], and a C-terminal cysteine-rich region potentially involved in nucleic acid binding[24–27]. ORF1 and ORF2 preferentially bind the L1 mRNA in *cis* to form a ribonucleoprotein particle (RNP)[28]. The L1 RNPs contain many copies of ORF1 and only one or two copies of ORF2, but the stoichiometry needed for retrotransposition is unknown[29]. To insert a new copy into the genome, L1 uses target-primed reverse transcription (TPRT)[2]. The EN initiates this pathway by creating a single-stranded nick in genomic DNA at the semi-specific 5'-TTTT*A-3' consensus sequence[30], with the

[1]Department of Molecular Biology, Cell Biology, and Biochemistry, Brown University, Providence, RI, USA. [2]Center on the Biology of Aging, Brown University, Providence, RI, USA. ✉e-mail: gerwald_jogl@brown.edu; john_sedivy@brown.edu

asterisk marking the location of the cleaved phosphodiester bond. The exposed poly-T sequence base-pairs with the poly-A tail of the L1 transcript to prime reverse transcription by the RT to create L1 complementary DNA (cDNA). This step is followed by polymerization of the second L1 strand using a primer possibly generated by EN nicking. Finally, host factors integrate the new copy of L1, resulting in flanking target site duplications, a hallmark of canonical TPRT. Both the EN and RT are required for retrotransposition, as active site mutations abolish retrotranposition[20,22,31], though EN-independent retrotransposition can occur in cells deficient for non-homologous end joining and at dysfunctional telomeres[32,33]. Therefore, the EN plays a key role in beginning the process of creating and integrating a new L1 copy into the genome.

Previous studies have shown that pharmacological inhibition of the L1 RT by inhibitors originally designed for the Human Immunodeficiency Virus (HIV) RT reduces retrotransposition to a similar extent as active site mutation[34,35]. These inhibitors also decrease the amount of pro-inflammatory L1 cDNA found in the cytoplasm of senescent cells, as well as inflammation in aged mice[16,18]. However, no small molecule EN inhibitors have been characterized, although the EN is a promising target for several reasons. The crystal structures of the EN alone[36] and bound to substrate DNA[37] have been solved, enabling use of in silico screening of candidate small molecules. A similar approach was used to successfully identify inhibitors of apurinic/apyrimidinic endonuclease 1 (APE1)[38,39], a structurally related enzyme required for the base excision repair pathway[36]. The EN can also be easily produced in *E. coli* and therefore is amenable to biochemical characterization and in vitro inhibitor screening. In addition to these technical advantages, the DNA damage induced by L1 activity in cells is driven in part by the EN. Expression of full-length L1[10,12], ORF2 without ORF1[11,13,40], or the EN alone[11,41], has been shown to result in double-strand DNA breaks as measured by increased expression of phosphorylated histone H2AX (γ-H2AX), accumulation of p53-binding protein 1 (53BP1) foci, and nuclear fragmentation in the neutral comet assay. This DNA damage is impaired by mutation of the EN[10,11,41] and even more so when both the RT and EN are mutated[11,12,40]. Expression of ectopic full-length and truncated L1 elements can also reduce cell viability, which is partially rescued when EN is mutated, in some cases to a larger extent than RT mutation alone[10,11,14,41]. These results indicate that not only does L1 create DNA damage and resulting cytotoxicity, but that both can occur in the absence of retrotransposition and appear to be largely due to EN activity.

It has been suggested that up to 10 times more double-strand breaks occur in cells overexpressing L1 than productive insertions, based on relative frequencies of γ-H2AX foci and retrotransposition events[10]. EN inhibitors would be very useful to understand EN function in the context of natural L1 life cycles, since our current knowledge is mostly based on studies using ectopically introduced L1 overexpression constructs. Selective inhibition of the EN in senescent cells could also help elucidate the mechanism of cytoplasmic L1 cDNA formation and subsequent triggering of the IFN-I response. These inhibitors would therefore complement existing RT inhibitors to assess the relative contributions of both enzymatic ORF2 domains to L1-induced phenotypes. Finally, development of EN inhibitors would enable combining pharmacological inhibition of both the RT and EN for testing additive or synergistic therapeutic effects.

Here we describe a structurally diverse set of small molecule EN inhibitors. We have identified these compounds through computational screening methods and quantified their respective efficacies biochemically and biophysically. We have also shown that these inhibitors impact multiple aspects of the L1 life cycle: retrotransposition, L1-induced DNA damage, and expression of inflammatory factors in senescent cells. These results present evidence that EN inhibition can reduce the expression of inflammatory factors in senescent cells. These inhibitors can be used as tools for better understanding L1

function from a basic science perspective and as initial candidates for the development of therapeutics.

## Results

### Overview of EN inhibitor testing strategy

We screened potential EN inhibitors using several approaches in succession as shown in Fig. 1. We first used computational strategies to generate four groups of candidate compounds from multiple existing or filtered libraries. Next, we used several molecular docking programs to predict compounds with high affinity in order to choose compounds for in vitro testing. We used two assays to test inhibition of purified EN by these candidate compounds and measured direct EN and inhibitor binding. We also determined the effectiveness of the EN inhibitors in reducing L1 retrotransposition in cell culture, followed by testing of inhibitors with cellular activity against L1-induced DNA damage and expression of inflammatory markers in senescent cells.

To aid our docking efforts, we solved two EN structures by x-ray crystallography (Supplementary Fig. 1, Supplementary Table 1). The first was a structure of EN bound to a manganese ion ($Mn^{2+}$) (Supplementary Fig. 1a, PDB ID: 8SP5). Previous structures of the wild-type EN at the time this work was completed did not contain bound catalytic metal ions. Our structure is consistent with the recently published structure with a magnesium ion ($Mg^{2+}$) bound to the EN[37] and shows involvement of residues that are mostly conserved with the coordinating residues in various APE1 structures containing metal ions[42,43]. Since the EN requires either $Mg^{2+}$ or $Mn^{2+}$ for activity, we believe this provides a better representation of the electrostatic environment of the active site for docking than the apo structure (protein without a metal ion). The second structure is of the EN bound to a low molecular weight compound, tranexamic acid (Supplementary Fig. 1b, PDB ID: 8SP7), which we discovered through fragment screening by crystallography[44]. This provided empirical structural information about binding of a small molecule to inform docking, as described in more detail below.

### Identification of a preliminary EN inhibitor and in silico screening

Before evaluating large libraries of compounds, we sought an initial inhibitor to serve as a starting point, as no EN inhibitors have been previously characterized. We thus explored inhibitors of the cellular enzyme APE1[38], which is structurally similar to the EN[36]. A similar approach was used to discover that some HIV RT inhibitors, such as the widely-used drug Lamivudine (also known as 3TC), have efficacy against the L1 RT due to similarities in sequence and enzymatic activity, and can therefore be repurposed for L1 inhibition[34,35]. While the active sites of both the EN and APE1 contain conserved residues that might interact similarly with compounds, the binding surfaces adjacent to the DNA cleavage site differ significantly (area to the right of H230, see Supplementary Fig. 2) due to their diverging substrates and functions[36]. For example, the structurally analogous site to the EN S202 is the APE1 W280 (Supplementary Fig. 2a), which decreases the size of the APE1 binding surface relative to the EN (Supplementary Fig. 2b). In our molecular docking experiments, we included the full EN DNA binding site (area used for docking is shown in Supplementary Fig. 1a, Supplementary Fig. 2) to maximize potential interactions with candidate compounds and opportunities for specificity relative to APE1.

We started our in silico studies by using the docking program LeDock[45] and the crystal structure of the L1 EN[36] (PDB ID: 1VYB) to evaluate 15 reported APE1 inhibitors. We chose to test further one of the weaker APE1 inhibitors, NSC89640, which we refer to here as AD2 ($K_i$ for APE1 = 13 μM[38]), as it had the most favorable docking energy with EN. The APE1 inhibitor with the best efficacy against APE1, NSC332395, which we refer to here as AD1 ($K_i$ for APE1 = 0.12 μM[38]), was not ranked as well by docking with EN. While interaction of AD1 and APE1 has been shown by a fluorescence binding assay[38], direct binding

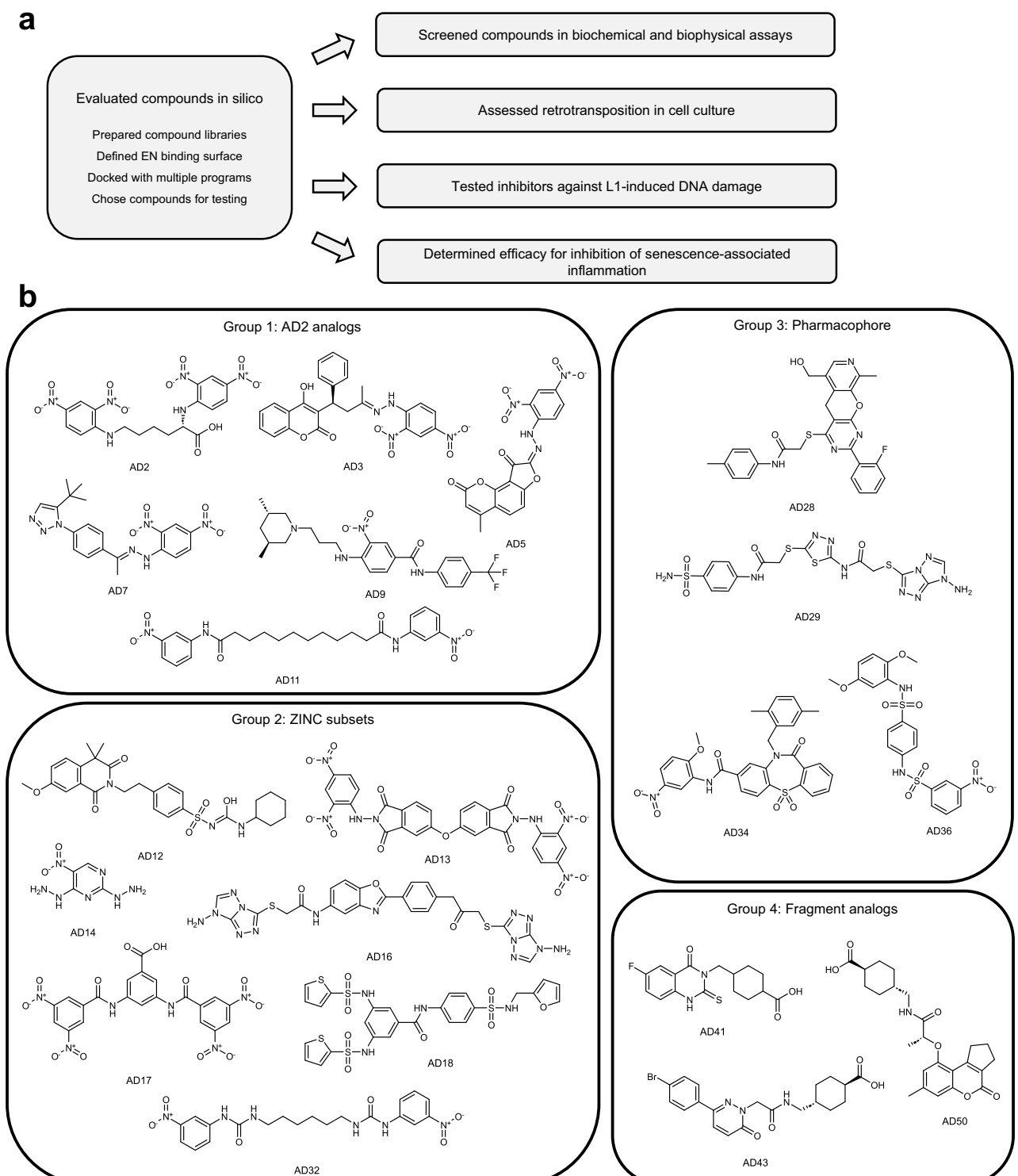

**Fig. 1 | Workflow for identifying and testing EN inhibitors. a** Schematic of the workflow. Programs used for docking: LeDock, AutoDock Vina, DOCK 6.9, FitDock. Receptors used for docking: apo EN structure (PDB ID: 1VYB), EN/Mn$^{2+}$ structure (PDB ID: 8SP5). The EN/Mn$^{2+}$ structure was used for docking all groups shown here except AD2 analogs. **b** Structures of EN inhibitors described in this study are shown arranged in docking Groups 1–4 based on the strategies used to identify them. Small molecule structures were generated in ChemDraw. For a detailed description of the workflow, see the Methods section.

of AD2 to APE1 has not been evaluated, so the specific interactions of this compound with APE1 are unknown. We evaluated these two preliminary candidate compounds using an established plasmid nicking assay for EN activity[22]. We found that AD1 did not inhibit EN activity in this assay, whereas AD2 resulted in full inhibition at 1 mM (Supplementary Fig. 3).

While AD2 showed only weak inhibition in this activity assay, we used this compound as a starting point for our first round of docking. We selected structural analogs of AD2 identified by the ZINC database[46], and evaluated them using LeDock[45], AutoDock Vina[47], and DOCK 6.9[48]. In using multiple algorithms for screening, we aimed to increase the accuracy of our predictions by choosing compounds with

favorable energies and/or similar binding positions in the designated DNA binding surface containing catalytic residues (Supplementary Fig. 4). The inhibitors identified from this group were AD3, AD5, AD7, AD9, and AD11 (Group 1, Fig. 1). Second, we expanded the structural diversity of the compounds screened by using the structure of EN bound to $Mn^{2+}$ instead of the apo structure, and were able to find additional inhibitors using LeDock and AutoDock Vina by screening ZINC subsets, including FDA and internationally approved drugs. The inhibitors characterized from this group were AD12, AD13, AD14, AD16, AD17, AD18, and AD32 (Group 2, Fig. 1). Our previous attempts to screen compound libraries unbiased by AD2 used the apo EN structure and resulted in no compounds with in vitro efficacy. Therefore, we used the structure of the EN bound to $Mn^{2+}$ for docking of Groups 2, 3, and 4 (Fig. 1).

Third, we used additional approaches incorporating structural information to better inform our docking strategies. We generated a structure-based pharmacophore using a structure of the EN bound to substrate DNA[37] (PDB ID: 7N94). We used ZINCPharmer to generate the pharmacophore and filter the ZINC database for compounds that matched the pharmacophore, followed by docking with LeDock and AutoDock Vina. Such a pharmacophore-based approach was also used to filter candidate APE1 inhibitors[39]. The inhibitors identified in this manner were AD28, AD29, AD34, and AD36 (Group 3, Fig. 1). Fourth, we used the structure of the EN bound to the fragment tranexamic acid (PDB ID: 8SP7) identified by fragment screening (Supplementary Fig. 1b, see above) as yet another structure-based strategy that begins with empirically determined binding of a small molecule rather than computational predictions alone. In this approach, we screened analogs of this fragment using structural similarity and substructure searches. This provided a series of compounds with minor modifications to the fragment, while others were expanded into larger compounds more likely to inhibit activity. To choose compounds to test in vitro we used template docking with FitDock[49], in which analogs were overlaid with the fragment's location in the EN active site to serve as the initial location from which the docking calculations were completed. The inhibitors chosen from this group were AD41, AD43, and AD50 (Group 4, Fig. 1).

## Fluorescent oligonucleotide activity assay to quantify EN inhibitor efficacy

After docking, we used two assays to biochemically characterize candidate EN inhibitors. We initially tested AD2 analogs and some compounds from unbiased ZINC subsets with the plasmid nicking assay, and found several compounds with inhibition (Supplementary Fig. 3). While this assay provided qualitative evidence for inhibition, we sought a more quantitative assay in order to prioritize inhibitors for future development and testing in cells. Therefore, we adapted a fluorescent hairpin oligonucleotide assay previously used for APE1[38] for use with the EN (Supplementary Fig. 5). We replaced the abasic site analog and surrounding sequence with the EN target sequence 5'-TTTTA-3' (Fig. 2a). EN activity releases the fluorescently-tagged sequence that dissociates from the remaining sequence containing the quencher. This allows for real-time monitoring of activity based on fluorescence intensity and quantification of initial reaction rates across multiple inhibitor concentrations. We first tested inhibitors with confirmed activity in the plasmid assay in the fluorescent oligonucleotide assay to better quantify relative efficacies. Then, we utilized this assay to screen subsequent rounds of inhibitors. By measuring activity under multiple turnover conditions, we calculated $IC_{50}$ values by non-linear fit (Fig. 2b). We have replicated these results in at least triplicate and the average $IC_{50}$ values are reported in Table 1. Importantly, we obtained multiple inhibitors from each of the four docking groups (Fig. 1) with efficacy in this assay. Therefore, we have biochemically characterized EN inhibitors with diverse structural scaffolds and potency down to the low micromolar range.

To assess whether our inhibitors might also inhibit APE1, we performed the fluorescent oligonucleotide assay with a substrate containing an abasic site as previously described[38], and APE1 enzyme purchased from New England Biolabs. No EN inhibitors reduced activity to the same extent as AD1 and for the most part only minimal inhibition of APE1 by EN inhibitors occurred, if any (Supplementary Fig. 6a). We also tested inhibitors soluble at 250 µM (AD2, AD3, AD7, AD9, AD12, AD32, AD36, AD43), the concentration at which inhibition by the weak APE1 inhibitor AD2 was detected (Supplementary Fig. 6b). These EN inhibitors resulted in no activity at this concentration, showing that they are even less potent against APE1 than AD2. These results are consistent with the differences in the DNA binding surfaces between the two enzymes (Supplementary Fig. 2), and the docked locations of EN inhibitors within these areas (Supplementary Fig. 4); for example, many EN inhibitors, but not AD1, are predicted to interact with S202, which in APE1 is replaced by W280.

## Biophysical characterization of EN inhibitors

After quantifying the inhibition of in vitro EN enzymatic activity by our compounds, we sought to measure direct interactions between the inhibitors and the EN in the absense of DNA. We used the spectral shift method[50], in which fluorescently-labeled EN is incubated with varying concentrations of inhibitor. Changes in the chemical environment of the fluorophore, including nearby binding of a ligand or conformational changes in the protein, cause a shift in the emission spectrum. By calculating the ratio of fluorescence at two wavelengths as a function of inhibitor concentration, a binding curve can be generated and a dissociation constant ($K_d$) fit applied. Since this shift can either be a "red" shift or "blue" shift, the ratio value of the bound state may be higher or lower than the unbound state, resulting in a curve that may increase or decrease with higher inhibitor concentrations (Fig. 3). As with the oligonucleotide assay, we have completed these measurements in at least triplicate and the average $K_d$ values are shown in Table 1. Of the nine compounds with $IC_{50}$ values lower than 100 µM, five had $K_d$ values below their $IC_{50}$ values (ranging from 1.1-fold to 11.5-fold lower; AD5, AD13, AD17, AD18, AD34), and four had $K_d$ values above their $IC_{50}$ values (ranging from 1.6-fold to 4.8-fold higher; AD11, AD14, AD16, AD29) (Table 1). Two compounds (AD9, AD28) had $IC_{50}$ values greater than 100 µM but $K_d$ values in the low µM range (30–80-fold differences). We currently do not understand the reasons for these differences; one possibility is that some compounds might interact with regions on EN outside the DNA-binding site, resulting in good $K_d$ values but poor inhibition of enzymatic activity. We also confirmed that the inhibitors do not show significant binding to the oligonucleotide substrate (Supplementary Table 2).

## EN inhibitors reduce L1 retrotransposition in cell culture

After characterizing EN inhibitors biochemically and biophysically, we examined their effects on cellular consequences of L1 activity. We first evaluated L1 retrotransposition, as this represents the most direct and well-characterized biological L1 activity. We used an established dual-luciferase reporter assay in HeLa cells to measure L1 retrotransposition[20,51]. In this assay, expression of a Firefly luciferase reporter occurs only after completion of a full L1 life cycle, beginning with L1 mRNA expression and ending with insertion of a new L1 element in the genome. We found that several compounds with in vitro efficacy reduced retrotransposition (Fig. 4a). Significantly, we found at least one inhibitor from each docking group (Fig. 1) with cellular activity. The RT inhibitor 3TC served as a measure of inhibition the RT domain, which is well documented to be required for retrotransposition[20]. Several EN inhibitors decreased retrotransposition to a similar degree as 3TC, while for others inhibition was less potent. We also monitored overall cellular

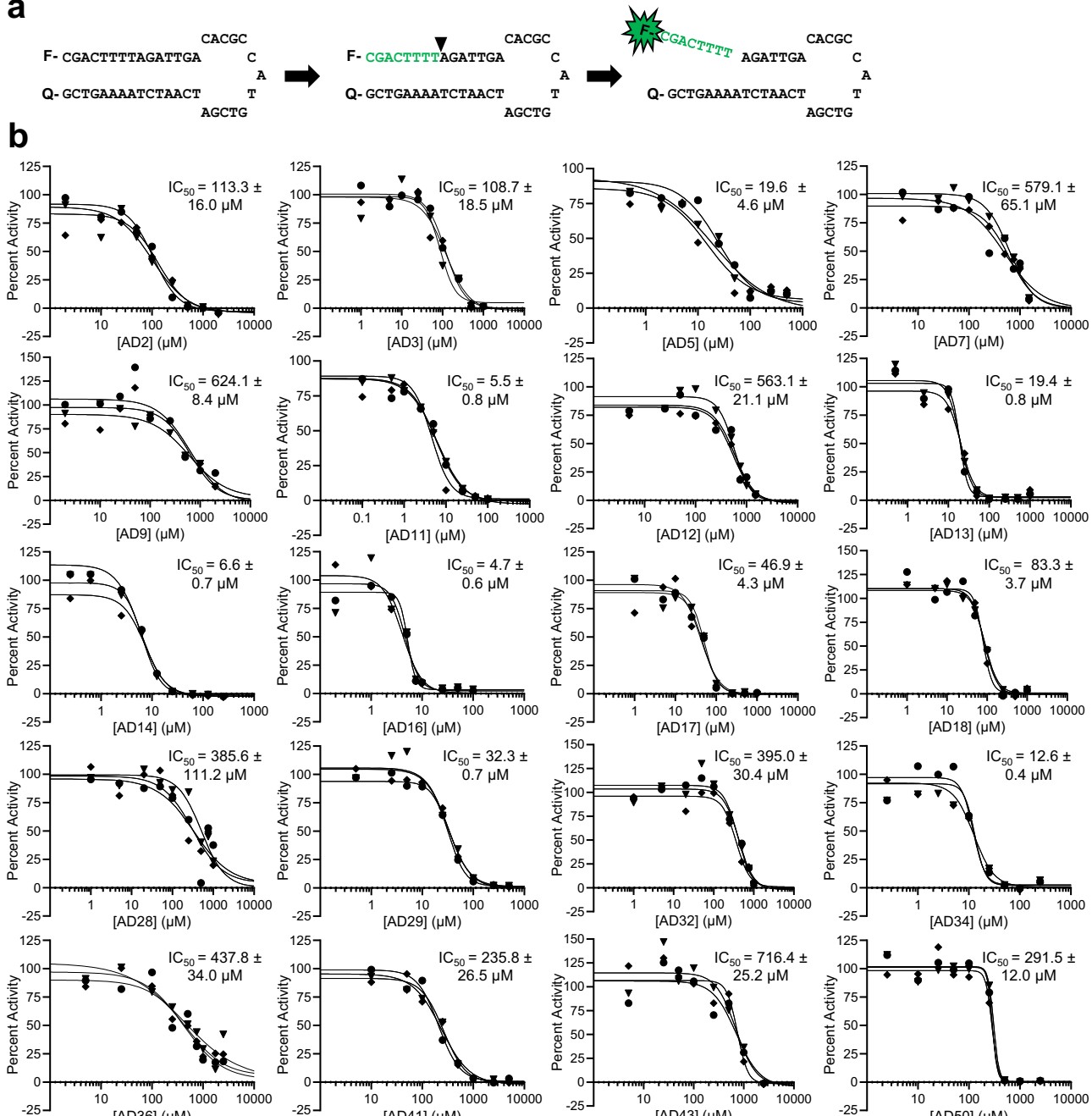

**Fig. 2 | Inhibition by EN inhibitors determined with a fluorescent oligonucleotide activity assay. a** Schematic of the fluorescent oligonucleotide assay. Left: sequence of hairpin oligonucleotide containing 5′ 6-FAM fluorescein fluorophore (F) and 3′ DABCYL quencher (Q). Middle: arrowhead indicates location of nick by EN at the semi-specific target site sequence 5′-TTTT*A-3′. The melting temperature of the green sequence is lower than the reaction temperature, whereas the melting temperature of the full hairpin is higher than the reaction temperature. Right: the fluorescently-tagged nicked sequence (8 nucleotides) is released from the quencher sequence (35 nucleotides), allowing for fluorescence to occur as a real-time readout of EN activity. **b** Representative assay results for each EN inhibitor.

Activity was determined as the initial rate of reaction under multiple turnover conditions. Graphs show the percent of no inhibitor control as a function of indicated inhibitor concentration. Average $IC_{50}$ values ± s.d. were calculated using the four parameter [inhibitor] vs. response non-linear fit in GraphPad Prism for each of the 3 technical replicates (shown as circles, diamonds, and triangles along with corresponding curve fits) in order to obtain s.d. values. No inhibitor control and full inhibition by 50 mM EDTA were included in fit calculations to guide definition of top and bottom of curve fits. Average $IC_{50}$ values calculated across at least 3 independent experiments can be found in Table 1. Source data are provided as a Source Data file.

toxicity using the PrestoBlue viability assay (Invitrogen), as previously used in conjunction with retrotransposition assays[52]. Dose-responsive inhibition was observed for several of our compounds (Fig. 4b). A summary of the average retrotransposition efficiencies calculated from at least three independent experiments is shown in Table 1.

**L1-induced DNA damage is reduced by EN inhibitors**

We next tested the effects of EN inhibitors on DNA damage associated with L1 expression. Inactivation of EN by an active site point mutation has been previously shown to significantly reduce DNA damage when L1 is overexpressed[10,11,41]. HeLa Tet-On cells were transfected with doxycycline-inducible plasmid constructs expressing full-length active

**Table 1 | Summary of EN inhibitor IC$_{50}$, $K_d$, and retrotransposition efficiency values**

| Name | IC$_{50}$ (μM) | $K_d$ (μM) | Retrotransposition efficiency (%) |
|---|---|---|---|
| AD2 | 104.0 ± 8.8 | 82.9 ± 34.5 | 50 μM: 98.0 ± 22.9 |
| AD3 | 114.4 ± 22.7 | 11.9 ± 7.9 | 20 μM: 59.0 ± 22.4 |
| AD5 | 16.1 ± 5.4 | 1.4 ± 0.2 | 10 μM: 89.7 ± 14.0 |
| AD7 | 866.4 ± 249.3 | 230.4 ± 60.2 | 20 μM: 42.2 ± 23.9 |
| AD9 | 468.2 ± 117.8 | 5.9 ± 2.5 | 50 μM: 21.9 ± 10.9 |
| AD11 | 4.0 ± 1.2 | 11.3 ± 7.4 | 50 μM: 52.2 ± 14.4 |
| AD12 | 671.2 ± 139.6 | 755.7 ± 398.7 | 50 μM: 17.7 ± 1.9 |
| AD13 | 25.0 ± 4.9 | 22.5 ± 11.2 | 20 μM: 75.4 ± 30.7 |
| AD14 | 6.0 ± 1.7 | 23.0 ± 9.5 | 25 μM: 71.6 ± 10.4 |
| AD16 | 5.8 ± 1.0 | 9.6 ± 1.8 | 25 μM: 59.2 ± 21.7 |
| AD17 | 46.6 ± 7.3 | 41.4 ± 13.0 | 5 μM: 94.8 ± 15.8 |
| AD18 | 76.3 ± 8.5 | 6.8 ± 2.8 | 10 μM: 92.2 ± 16.4 |
| AD28 | 420.7 ± 30.9 | 14.7 ± 3.7 | Toxicity at 2.5 μM |
| AD29 | 28.8 ± 3.2 | 139.5 ± 42.2 | 50 μM: 85.8 ± 7.9 |
| AD32 | 458.7 ± 82.6 | N.D. | 25 μM: 35.1 ± 16.5 |
| AD34 | 12.5 ± 4.2 | 1.8 ± 0.4 | 25 μM: 104.3 ± 7.8 |
| AD36 | 865.7 ± 300.1 | 242.9 ± 36.9 | 25 μM: 42.9 ± 22.2 |
| AD41 | 514.2 ± 156.1 | N.D. | 25 μM: 129.4 ± 12.0 |
| AD43 | 722.2 ± 135.3 | 516.4 ± 44.0 | 50 μM: 57.3 ± 9.0 |
| AD50 | 290.6 ± 13.0 | 207.6 ± 42.7 | 25 μM: 100.5 ± 15.0 |

IC$_{50}$ values were calculated from at least 3 independent experiments of the fluorescent oligonucleotide nicking assay and are shown as mean ± s.d. $K_d$ values were calculated from at least 3 independent spectral shift experiments and are shown as mean ± s.d. N.D., binding not detected. Retrotransposition efficiencies calculated as percent of the no inhibitor control were obtained from at least 3 independent experiments and are shown as mean ± s.d. Individual values used to calculate mean ± s.d values shown here can be found in the Source Data file.

L1 (FL, containing both ORF1 and ORF2) or the EN domain only. In addition to enzymatically active (WT) constructs, we used versions with the catalytically-dead mutations for EN, RT, or both. The impacts of EN inhibitors on L1-induced DNA damage were assessed with two assays: γ-H2AX immunofluorescence staining and the neutral comet assay. The conversion of histone H2A into its phosphorylated form, known as γ-H2AX, is triggered by double-strand DNA breaks (DSB) and is a well-known and often used cellular assay for DNA damage[53] (Supplementary Fig. 7a, Supplementary Fig. 7b). The neutral comet assay detects overall DNA fragmentation by exposing cell nuclei to electrophoresis and measuring the amount of DNA that migrates out of the nucleus[54] (Supplementary Fig. 7c). Mean γ-H2AX signals for each nucleus were quantified using the CellProfiler[55] software (Fig. 5a, b). OpenComet[56] software was used to identify nuclei and measure the comet parameter of tail length (Fig. 5c). Point mutations in RT and/or EN domains either prevented or significantly reduced DNA damage, consistent with previously published results. Our results show that the EN inhibitors reduce γ-H2AX signal both when FL L1 is expressed, as well as when the EN domain is expressed alone. We also included treatment with etoposide (Supplementary Fig. 7b) or hydrogen peroxide (Supplementary Fig. 7c) as controls for DNA damage. In both assays, we observed decreases in DNA damage in response to treatment with several inhibitors, with the amount of reduction falling between the uninduced and WT samples. Together, these assays demonstrate that small molecule EN inhibitors can reduce DNA damage caused by the expression of L1.

**EN inhibitors impact senescence-associated inflammatory markers**

The upregulation of interferon and inflammatory markers by L1 in senescent cells is of particular interest because, contrary to retrotransposition and DNA damage, it is believed to be triggered by cytoplasmic L1 cDNA sequences[16,27]. To evaluate this cellular impact of L1 expression, we generated replicatively senescent cells in culture and treated them with our EN inhibitors. Human diploid fibroblasts (LF1 cell line[57]) were passaged until they were no longer dividing, then these senescent cultures were maintained for 3–5 months before treatment with inhibitors and sample processing (Fig. 6a). These durations were based on previous evidence from our laboratory demonstrating significant expression of L1 at later stages of senescence beginning around 3 months and remaining consistent as cells are maintained in culture[16]. We also performed a variety of assays to confirm senescence as previously described[16], including tracking growth rate, monitoring cellular morphology, performing the senescence-associated β-galactosidase assay[58], measuring expression of interferon and inflammatory factors (IL6, IL1β, CCL2, IFNα) and senescence markers (p16, MMP3), and visualizing the presence of γ-H2AX foci, ORF1 protein, and cytoplasmic DNA/RNA hybrids by immunofluorescence (Supplementary Fig. 8).

We then used these cultures to test the effects of EN inhibitors in senescent cells. After treating cells with inhibitors for 1 month, we extracted the RNA and measured levels of markers of the pro-inflammatory SASP and IFN-I response by quantitative reverse transcription PCR (RT-qPCR) (Fig. 6b). In all experiments we included 3TC as a control for RT inhibition. We found mostly similar effects for 3TC and EN inhibitors on the indicated markers across three independent cultures of senescent cells. In order to assay additional gene expression changes upon inhibitor treatment, we performed RNA sequencing (RNA-seq) transcriptomic analysis (Fig. 6c). We found that 3TC and AD12 both resulted in decreased expression of our previously published IFN-I gene set[16] and a set of genes associated with aging[59]. These results demonstrate that pharmacological inhibition of both the RT and EN domains mitigates expression of inflammatory factors in senescent cells.

## Discussion

We describe here a set of structurally diverse small molecule L1 EN inhibitors. We identified these inhibitors by computational screening methods and quantified their efficacies with the purified EN domain. Furthermore, we have demonstrated that these inhibitors also mitigate retrotransposition and L1-induced DNA damage in HeLa cells, and the expression of interferon and inflammatory markers in senescent fibroblasts, all of which are disease-relevant impacts of L1 activity. We anticipate that these inhibitors will be useful tools for studying the basic biology of L1, and will serve as initial candidates for the development of therapeutics for L1-associated diseases.

We found differential potency among the inhibitors when comparing results from the in vitro and cellular assays. Several inhibitors showed activity in vitro (IC$_{50}$ and $K_d$ values near or below 100 μM) but did not significantly impact retrotransposition in cells: AD2, AD5, AD17, AD18, and AD34. This could be due to several factors including poor cell permeability, sequestration, and/or degradation. Among these inhibitors, AD5 ($K_d$ 1.4 μM, IC$_{50}$ 16.1 μM) and AD34 ($K_d$ 1.8 μM, IC$_{50}$ 12.5 μM) could be good starting points for further development of compounds with improved cellular properties; AD18 ($K_d$ 6.8 μM, IC$_{50}$ 76.3 μM) could also be of interest because of its unique functional groups relative to other inhibitors. Conversely, several compounds are designated as weak EN inhibitors due to their low efficacies in one or both of the in vitro assays, while still showing some activity in the retrotransposition assay: AD7, AD9, AD12, AD32, AD36, and AD43. The most likely explanation is that these inhibitors are "off-target", i.e., they influence other processes in the cell, which in turn indirectly impact retrotransposition. Since some of these inhibitors also affected DNA damage and/or expression of inflammatory factors, more investigation might be of interest. Finally, several inhibitors showed efficacy across in vitro and in vivo assays: AD3, AD11, AD14, AD16, and AD29. Based on

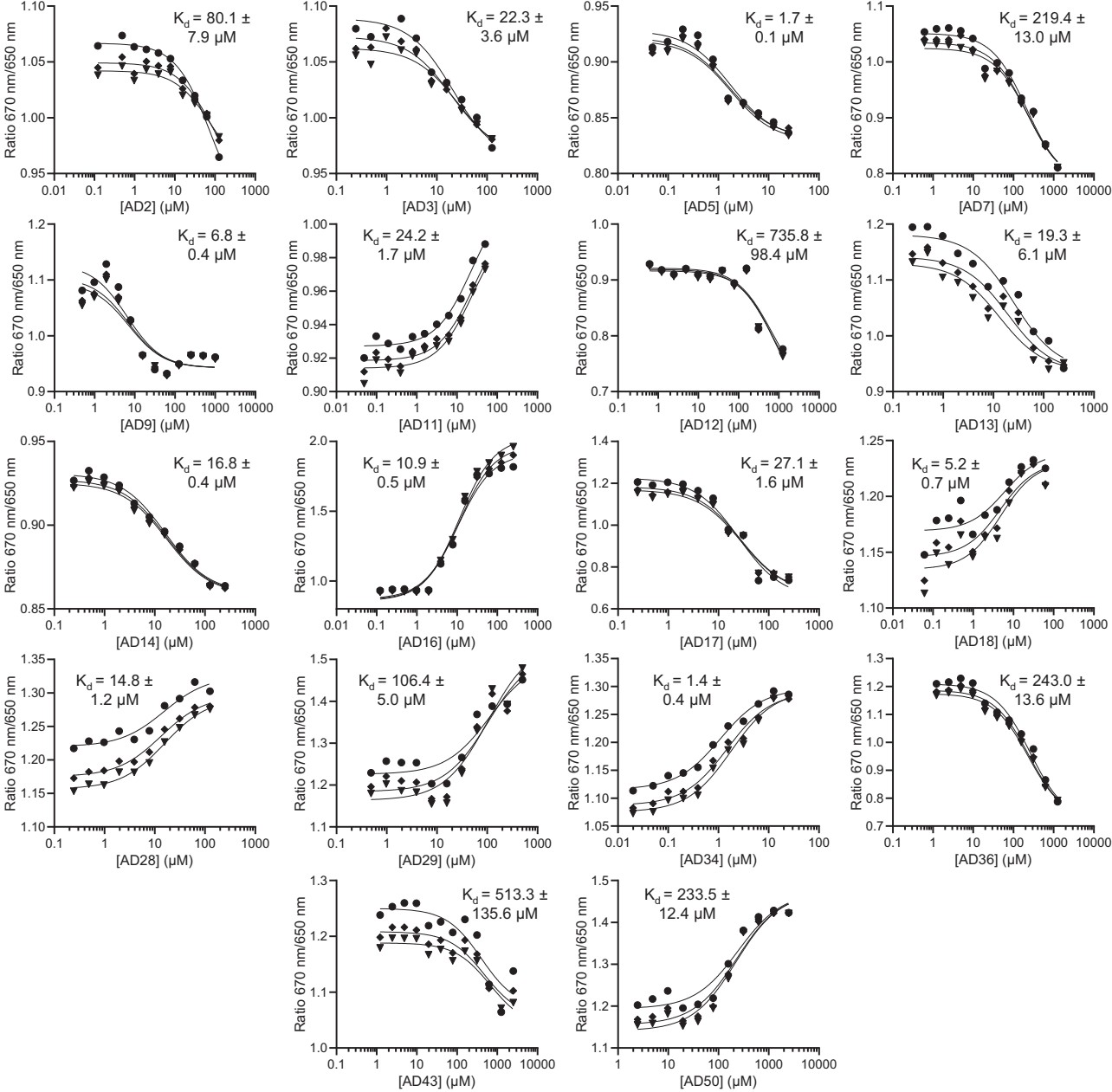

**Fig. 3 | Binding affinities of EN inhibitors determined with a spectral shift assay.** Representative assay results are shown for each EN inhibitor with detectable binding. EN was fluorescently tagged and incubated with indicated concentrations of inhibitors. Spectral shift measurements were performed on the Monolith X instrument (NanoTemper). Average $K_d$ values ± s.d were calculated in GraphPad Prism using the law of mass action as described[50] for each of the 3 technical replicates (shown as circles, diamonds, and triangles along with corresponding curve fits) in order to obtain s.d. values. Average $K_d$ values calculated across at least 3 independent experiments can be found in Table 1. Source data are provided as a Source Data file.

---

this consistency, impacts in multiple cellular assays (Figs. 4 and 6), and diversity of sizes and structures, these would be candidates for further exploration.

One possible explanation for discrepancies between in vitro and cellular assays is that some compounds might interact differently with the EN in the context of the full-length ORF2 versus the isolated domain, which was used for all in vitro work. Some evidence suggests that EN activity is reduced in full-length ORF2[60], potentially resulting from partial occlusion of the active site and therefore inhibitor binding surface. Conversely, the full-length ORF2 could provide additional binding surfaces for inhibitor interactions, or EN conformation and inhibitor binding could be different in ORF2 and the EN domain alone. Inhibitor binding to EN might also affect retrotransposition by impairing ORF2 function overall, including the RT domain directly or

the coordination between the domains during TPRT. Cellular EN activity occurs within a complex L1 RNP that interacts with multiple host proteins[29,61]. These interacting proteins vary depending on the localization of the L1 RNP in the nucleus or cytoplasm, and so could have different effects on the ability of the EN to bind to inhibitors. As a result, interactions between the EN and inhibitors may also be influenced by the step in the L1 life cycle.

While the retrotransposition assay requires full-length, intact L1, the DNA damage assays were performed using both full-length L1 and EN domain-only constructs. This allowed the evaluation of DNA damage inhibition in the context of retrotransposition-competent L1 elements in comparison to the EN domain alone. L1 elements that are incapable of retrotransposition but retain potentially active EN coding sequences are abundant in the human genome and thus have the

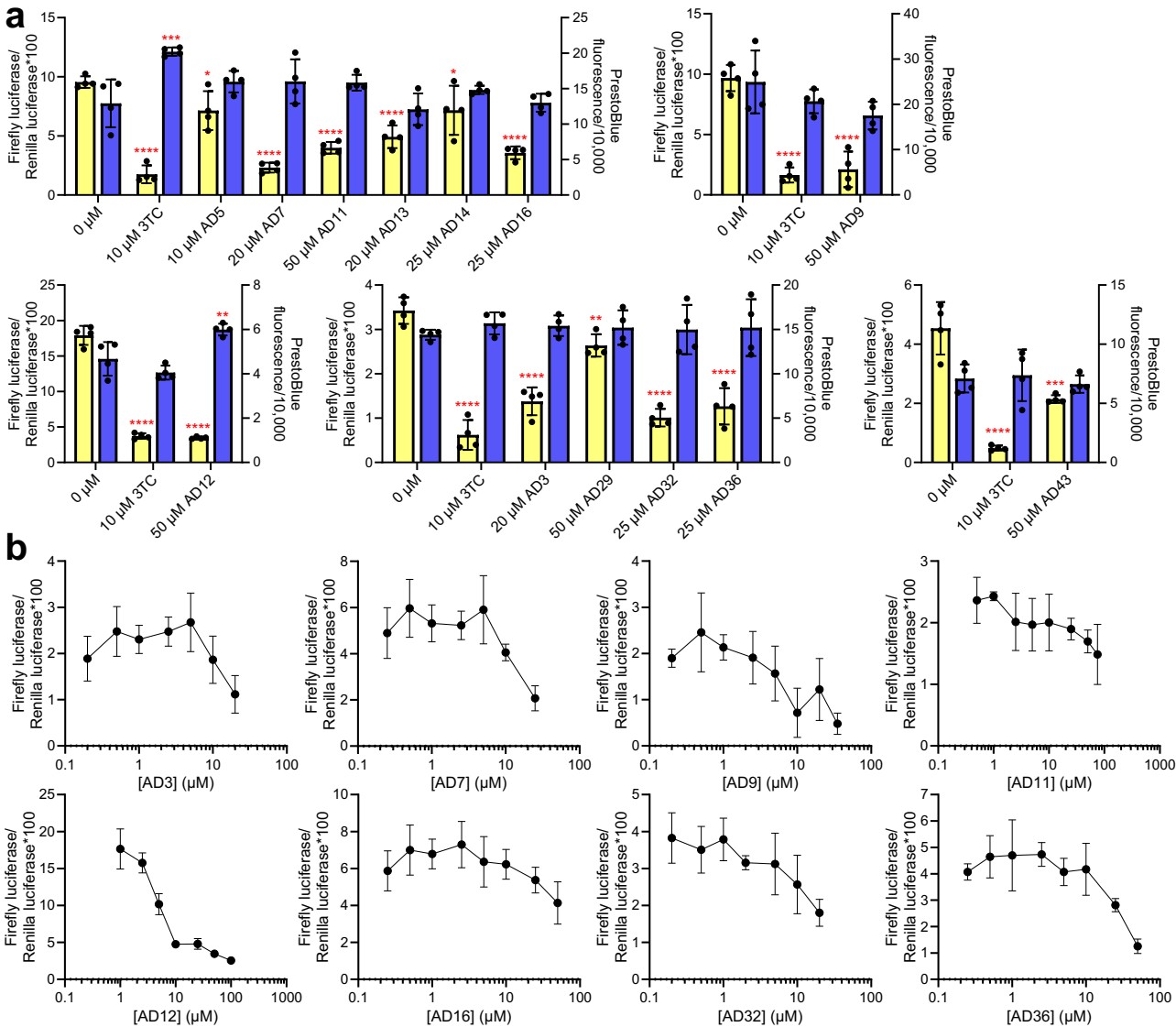

**Fig. 4 | Reduction of L1 retrotransposition in cell culture by EN inhibitors.**
**a** Representative retrotransposition assay results for EN inhibitors. Retrotransposition was assessed in HeLa cells containing a dual-luciferase L1 reporter construct (see Methods). Retrotransposition (y-axis, yellow bars) is shown as Firefly luciferase activity (measure of genome insertion) normalized to Renilla luciferase activity (measure of donor plasmid abundance). 3TC was included as a positive control for L1 inhibition. PrestoBlue Viability Reagent (y-axis, blue bars) was used to test cytotoxicity. Compounds with statistically significant cytotoxicity were subsequently tested at lower concentrations or excluded from further cellular testing. Statistical significance of the mean relative to no inhibitor control (0 μM) was determined by one-way ANOVA followed by Dunnett's multiple comparisons test using GraphPad Prism: *$p < 0.05$, **$p < 0.01$, ***$p < 0.001$, ****$p < 0.0001$. Data are mean ± s.d ($n = 4$ samples). Each graph represents an independent experiment. Average retrotransposition efficiency values calculated across at least 3 independent experiments can be found in Table 1. **b** Concentration-dependent inhibition for selected EN inhibitors. Results are from 4 replicates for each treatment and concentration, and presented as mean ± s.d ($n = 4$). Source data and exact $p$ values are provided as a Source Data file.

potential to create DNA damage if expressed[11]. It is interesting to note that inhibitors with the best inhibition in the full-length L1 DNA damage experiments were generally the ones with the best inhibition in the retrotransposition assay, for example AD3 and AD12. In a similar way, inhibitors with better efficacy in the EN domain-only DNA damage experiments were more likely to have better in vitro IC$_{50}$ and $K_d$ values, such as AD14 and AD16. However, testing of additional EN inhibitors with varying affinities is required to support this pattern. Inhibition of inflammation in senescent cells showed efficacy for inhibitors in both these groups: AD3, AD12, and AD14.

Our results regarding EN inhibitor activity in retrotransposition and DNA damage agree with previous research demonstrating that when L1 is overexpressed, EN active site mutations prevent retrotransposition and DNA damage. However, analogous experiments have not been performed to investigate the role of the EN in the production of inflammatory L1 cDNA found in the cytoplasm of senescent cells. This is because of technical challenges inherent in creating mutations in endogenous active L1s, which are found at multiple loci in the genome. RT inhibitors such as 3TC have helped answer these questions with regard to RT function[16,18], but the absence of EN inhibitors has prevented similar experiments to investigate the role of the EN. Our results suggest that the EN is at least partially involved in this process in senescent cells, as EN and RT inhibition resulted in similar decreases in expression of some inflammatory markers.

Our current understanding of the EN's function in the L1 life cycle is limited to nicking of nuclear DNA during TPRT. Hence, the potential involvement of the domain in initiating the production of L1 cDNA found in the cytoplasm of senescent cells is a novel finding that requires further investigation. One possibility is that nuclear envelope

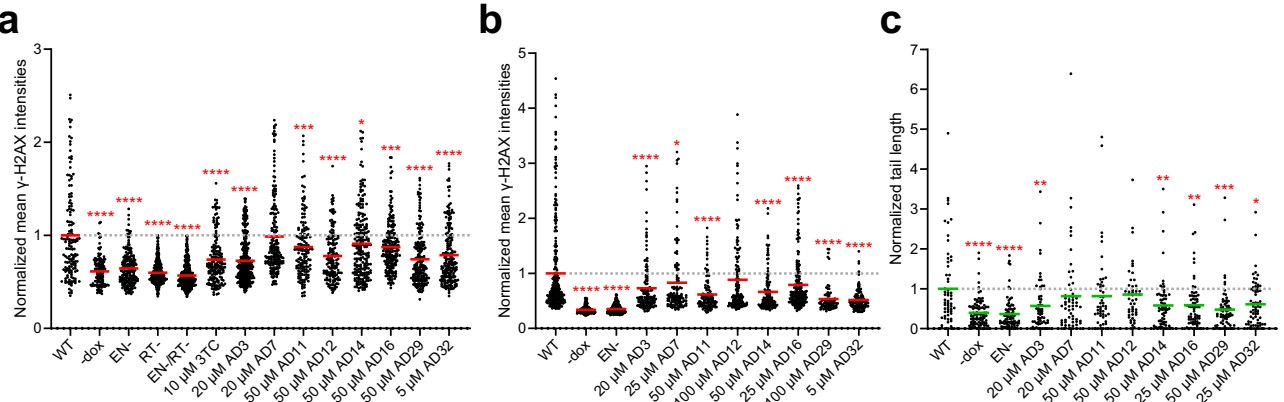

**Fig. 5 | Mitigation of L1-induced DNA damage by EN inhibitors.** HeLa Tet-On cells were transfected with doxycycline-inducible full-length L1 (FL), or EN domain-only expression constructs. In addition to active L1 (WT), we used constructs with catalytically-dead mutations for EN (H230A), RT (D702Y), and both EN and RT (H230A/D702Y). **a** DNA damage assessed by induction of γ-H2AX for FL constructs. Mean γ-H2AX intensities of individual nuclei were scored by immunofluorescence (see Methods); each dot represents one nucleus. The mean of the WT no inhibitor treatment was set to 1 (gray dotted line). Number of nuclei (n) per treatment: WT, $n = 143$; -dox, $n = 130$; EN-, $n = 197$; RT-, $n = 325$; EN-/RT-, $n = 413$; 3TC, $n = 173$; AD3, $n = 336$; AD7, $n = 197$; AD11, $n = 151$; AD12, $n = 155$; AD14, $n = 215$; AD16, $n = 220$; AD29, $n = 199$; AD32, $n = 195$. **b** DNA damage assessed by induction of γ-H2AX for EN domain-only constructs. Assay was performed as in (**a**) above. Number of nuclei (n)

per treatment: WT, $n = 301$; -dox, $n = 200$; EN-, $n = 363$; AD3, $n = 135$; AD7, $n = 117$; AD11, $n = 109$; AD12, $n = 142$; AD14, $n = 116$; AD16, $n = 204$; AD29, $n = 75$; AD32, $n = 111$. **c** DNA damage assessed with the neutral comet assay for EN domain-only constructs. Comet tail lengths were scored as indicated in Methods; each dot represents one comet tail from one nucleus. Number of nuclei (n) per treatment: WT, $n = 62$; -dox, $n = 90$; EN-, $n = 82$; AD3, $n = 59$; AD7, $n = 59$; AD11, $n = 52$; AD12, $n = 47$; AD14, $n = 68$; AD16, $n = 67$; AD29, $n = 68$; AD32, $n = 66$. Statistical significance of the mean (red and green solid lines) vs. WT was determined by one-way ANOVA followed by Dunnett's multiple comparisons test using GraphPad Prism: *$p < 0.05$, **$p < 0.01$, ***$p < 0.001$, ****$p < 0.0001$. Summary of results across at least 3 independent experiments can be found in Supplementary Table 3. Source data and exact $p$ values are provided as a Source Data file.

damage documented in senescent cells[62,63] allows for intermediates of retrotransposition primed in the nucleus by canonical TPRT to enter the cytoplasm and trigger the IFN-I response through the cGAS-STING pathway[64]. Another possible explanation is that the EN can function in the cytoplasm to improve RT priming by nicking cytoplasmic chromatin fragments found in senescent cells[65,66] or other sequences such as mitochondrial DNA, with preference for AT-rich sequences that promote RT priming[67]. Some evidence suggests that many of these extranuclear sequences are double-stranded[68] and would therefore require nicking. Yet another possibility is that nuclear EN activity promotes DNA fragmentation and creation of these chromatin fragments as part of the persistent DNA damage present in senescent cells[69]. Accumulation of cytoplasmic DNA has also been observed in response to pharmacological DNA damage induction[68]. Much more work is required to elucidate the role(s) of L1 in the genesis of cytoplasmic DNA species, a key hallmark of senescent cells.

The diversity of EN inhibitors described here, in combination with the relatively large DNA binding surface of the EN, suggests that more classes of EN inhibitors are theoretically possible. This could mitigate concerns regarding cross-reactivity with other cellular enzymes in addition to APE1 and provide better management of off-target effects. As previously noted, most of what we know about L1 has been garnered using overexpression models. While useful, this approach needs to be complemented with biologically relevant experiments that explore the role of endogenous L1 elements. Our inhibitors and future, higher affinity inhibitors should be very useful to study EN function and the effects of EN inhibition in a wide range of contexts, cell lines, animal models, and diseases, such as in cancer and neurodegeneration. RT inhibitors have been successfully used to test the contribution of RT activity to various L1-associated phenotypes found in different diseases[13,16,18,34,35,70-72]. Current RT inhibitors with activity against L1 rely on chain termination as the mechanism of action. EN inhibitors provide an orthogonal pharmacological approach to L1 inhibition that might be useful to clarify off-target effects of RT inhibitors. Finally, combination therapy with RT and EN inhibitors might be advantageous in some situations, as the EN domain is directly responsible for DNA damage.

In summary, we have characterized a variety of small molecule inhibitors of the LINE-1 retrotransposon endonuclease domain. In the short term, these inhibitors can serve as tools to improve our understanding of L1 biology in a similar way to how compounds repurposed for RT inhibition have been used. Ultimately, these inhibitors represent a starting point for future development of potential therapeutics for diseases associated with L1 expression.

## Methods
### Materials
AD1/NSC332395 was obtained from the National Cancer Institute. The following EN inhibitor was obtained from Sigma: AD12/ZINC1482077 (CDS021537). The following EN inhibitors were obtained from the listed suppliers through Molport: AD2/ZINC89469886/NSC89640 (Alinda IBS-L0127235), AD3/ZINC100299612 (ChemBridge 5151622), AD5/ZINC100499350 (ChemDiv 1440-2881), AD7/ZINC254379081 (Enamine Z56821059), AD9/ZINC20677610 (ChemDiv K784-1448), AD11/ZINC12428901 (ChemDiv 8003-9274), AD13/ZINC8398444 (Specs AG-690/11231133), AD14/ZINC5758200 (Specs AC-907/25004307), AD16/ZINC33355084 (Vitas STK672667), AD17/ZINC101372673 (Vitas STK000838), AD18/ZINC96022289 (UkrOrgSynthesis PB56889488), AD28/ZINC9116296 (ChemDiv E544-0411), AD29/ZINC33355295 (ChemDiv 8008-0573), AD32/ZINC9056988 (TimTec ST002110), AD34/ZINC33356589 (Specs AQ-088/42014071), AD36/ZINC16215374 (ChemBridge 7937857), AD41/ZINC425300 (Life Chemicals F0916-6060), AD43/ZINC238900190 (Eximed EiM08-19659), AD50/ZINC238924061 (Vitas STK717800). HeLa Tet-On cells containing plasmid pPM404[52], which expresses the doxycycline-inducible L1 WT *ORFeus* sequence with a dual-luciferase reporter for use in the L1 retrotransposition assay, were a gift from the laboratory of Jef Boeke. The following plasmids used for DNA damage experiments in HeLa Tet-On cells were gifts from the laboratory of Kathleen Burns[12]: pDA007 (Addgene plasmid #131380), pDA025 (Addgene plasmid #131384), pDA027 (Addgene plasmid #131385), and pDA034 (Addgene plasmid #131386).

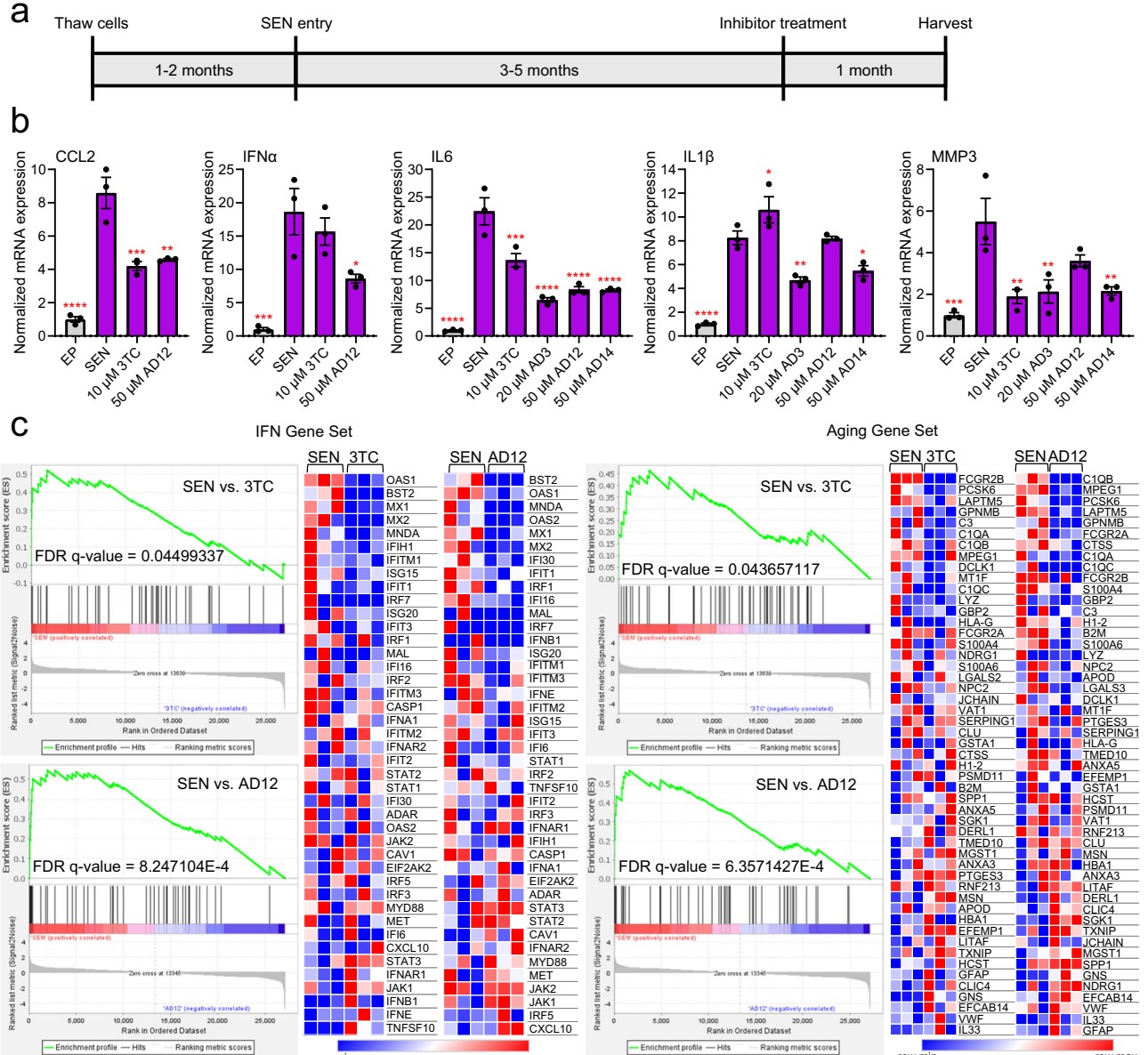

**Fig. 6 | Impact of EN inhibitors on inflammation markers in senescent cells.**
**a** Timeline of senescent (SEN) culture generation and treatment with inhibitors. Cells were passaged until they stopped dividing (SEN entry) and then maintained for 3–5 months, after which they were treated with inhibitors for 1 month before harvesting cultures for RNA. **b** RT-qPCR of inflammatory markers for 3 independent senescent cultures treated with inhibitors for 1 month beginning at the following time points: 4 months (CCL2 and IFNα), 5 months (IL6 and IL1β), or 3 months (MMP3) senescence. All senescent samples (purple bars) were normalized to early passage (EP, gray bars). Statistical significance of the mean relative to SEN (no inhibitor control) was determined by one-way ANOVA followed by Dunnett's multiple comparisons test using GraphPad Prism: *$p < 0.05$, **$p < 0.01$, ***$p < 0.001$, ****$p < 0.0001$. Data are mean ± s.e.m ($n = 3$ samples). **c** RNA-seq of senescent cultures treated for 1 month beginning at 3 months senescence. Gene Set Enrichment Analysis (GSEA) results for type-I interferon (IFN) and Aging gene sets. The order of genes in the gene expression (counts normalized in DESeq2) heat maps was determined by each gene's rank metric score as calculated by GSEA. The heat maps were generated in Morpheus (The Broad Institute). Source data and exact $p$ values are provided as a Source Data file.

## Cloning and plasmids

The untagged L1 EN WT protein expression plasmid was generated by restriction digest cloning. The L1 ORF2 consensus sequence was optimized for expression in *E. coli* and synthesized in pUC57 by GenScript. The sequence corresponding to residues 1-239 was amplified by PCR using Q5 High-Fidelity DNA Polymerase (NEB), cut with restriction enzymes NdeI (NEB) and XhoI (NEB), and ligated into digested pET26b with T4 DNA ligase (NEB). Individual colonies were tested by colony PCR and the insert was confirmed by sequencing following miniprep DNA (Qiagen). Plasmids expressing *ORFeus* EN domain only were generated by restriction digest cloning. The sequence corresponding to residues 1-239 from pDA007 (EN WT) or from pDA025 (EN H230A)

was amplified as described above, cut with BamHI (NEB) and PacI (NEB), and ligated into the digested pDA007 backbone as described above. Inserts were confirmed as described above. Primer sequences used can be found in Supplementary Data 1.

## Molecular docking and pharmacophore generation

The following programs were used for molecular docking: LeDock[45], AutoDock Vina[47], DOCK 6.9[48], and FitDock[49]. The published apo EN WT crystal structure (PDB ID: 1VYB) was used for docking APE1 inhibitors and AD2 analogs, while the EN/Mn$^{2+}$ structure (PDB ID: 8SP5) was used for all other docking experiments. The compounds were docked into the DNA binding site of the EN as determined based on active site

residues required for activity[22] and the predicted DNA interacting surfaced based on alignment with APE1[36] prior to the crystal structure of the EN bound to DNA[37] being published. All waters and ligands other than the $Mn^{2+}$ ion were removed from EN crystal structures prior to docking. For all programs, mol2 files were obtained from the ZINC database[46]. For AD2 analogs (429 compounds), the compounds were chosen by selecting "Find All" under AD2 "Interesting Analogs" and limiting to "For Sale" compounds. Existing ZINC subsets, such as all purchasable compounds approved by the FDA or similar international organizations ("world", 3278 compounds), or all purchasable compounds currently available ("in-stock", 12,084,317), were downloaded directly from ZINC and similarly limited to "For Sale". Due to the size of the "in-stock" subset, compounds were initially evaluated only with LeDock, then the top 20 to 100 compounds from each tranche were also evaluated with AutoDock Vina, resulting in 4185 compounds docked with both programs.

The pharmacophore was generated using the EN/DNA co-crystal structure[37] (PDB ID: 7N94) with ZINCPharmer[73]. Detected features were chosen to reflect key DNA and active site residue interactions and based on similarity to APE1/DNA pharmacophore features[39]. The final pharmacophore contained 5 features: 2 hydrophobic centers and 3 hydrogen-bond acceptors. The ZINC IDs for compounds matching the pharmacophore were provided by ZINCPharmer and then used to download the mol2 files for "For Sale" compounds from the ZINC database (1566 compounds). FitDock was used for template docking. The EN/tranexamic acid complex structure (PDB ID: 8SP7), (structure solution details below) was aligned with the EN/$Mn^{2+}$ structure in Fit-Dock to determine the fragment location within the metal-bound structure that was then used as the receptor for docking analogs of this fragment. Candidate analog compounds were found by similarity (0.6 cutoff) and substructure searches of the Molport catalog. The ZINC database was then searched by Molport ID to obtain the mol2 files as the input for docking (1959 compounds). The numbers of compounds reported reflect the contents of the ZINC database and Molport catalog at the times the searches were complete, rather than their updated current contents. Compounds from all docking groups were chosen for testing in vitro based on the default ranking by each program and agreement in binding positions among programs when applicable.

## EN expression and purification

The *E. coli* codon optimized EN plasmid was transformed into BL21 Star (DE3) competent cells (Invitrogen) for large-scale expression. EN cultures were grown at 37 °C in 50 μg/mL kanamycin until they reached an $OD_{600}$ of 0.6–0.9 and then were induced with 0.5 mM IPTG. The cultures were then grown for 2 h at 37 °C before harvesting by centrifugation at $4000 \times g$ for 12 min at 4 °C and then stored at −80 °C. Cell pellets were resuspended in 10 mL EN Lysis Buffer (20 mM HEPES pH 7.5, 300 mM NaCl, 1 mM DTT) for each 1 g of cell pellet and lysed with an Avestin EmulsiFlex C3 (ATA Scientific). Lysate was centrifuged at $100,000 \times g$ for 1 h at 4 °C and the supernatant was filtered prior to loading onto a manually packed 20 mL Heparin affinity column. Protein was eluted using a gradient of 30–100% Buffer B (Buffer A: 20 mM HEPES pH 7.5, 1 mM DTT; Buffer B: 20 mM HEPES pH 7.5, 1 M NaCl, 1 mM DTT), diluted to 400 mM NaCl, and loaded onto a manually packed Sepharose SP Fast Flow cation exchange column. Protein was eluted using a gradient of 40–100% Buffer B. Fractions containing protein were pooled, concentrated, filtered, and loaded onto a HiPrep Sephacryl S-100 16/60 size exclusion column. Protein was eluted, aliquoted, and stored at −80 °C in EN Lysis Buffer. Protein purification results were confirmed by SDS-PAGE.

## EN crystallization

EN crystals for the EN/$Mn^{2+}$ structure were grown by mixing an equal volume of 15 mg/mL protein in EN Lysis Buffer with an equal volume of crystallizing condition based on the published apo crystal structure:

0.14 M ammonium sulfate, 24% polyethylene glycol (PEG) 5000 monomethyl ether (MME), 5 mM magnesium chloride. Crystals were soaked in cryoprotecting solution containing crystallizing condition, 30% PEG 200, and 100 mM manganese sulfate before flash freezing in liquid nitrogen. Diffraction images were collected at the NSLS-II AMX 17-ID-1 beamline at Brookhaven National Laboratory at 100 K at a wavelength of 0.92 Å with the Eiger 9 M detector using the Life Science Data Collection software. Images were processed using XDS[74] and Aimless in CCP4[75]. The published apo structure[36] (PDB ID: 1VYB) was used as the search model for molecular replacement with Phaser in Phenix[76]. Anomalous diffraction maps were generated to determine the location of the $Mn^{2+}$ ion. The structure was finished by iterative rounds of manual building in Coot[77] and refinement in Phenix. $Mn^{2+}$ coordination by active site residues and water molecules was evaluated with CheckMyMetal[78].

Frag Xtal Screen (Jena Bioscience) was used for fragment screening by crystallography experiments, which yielded the EN/tranexamic acid structure. Crystals for fragment screening were grown by mixing an equal volume of 16.2 mg/mL protein in EN Lysis Buffer plus 10% DMSO with an equal volume of crystallizing condition: 0.1 M Tris acetate pH 6.0, 0.2 M lithium sulfate, 30% PEG 2000 MME. The tranexamic acid soak was performed for 2 h at room temperature in solution containing crystallizing condition, 50 mM tranexamic acid, 2.5% DMSO, 9.6% glycerol, and 10% ethylene glycol, before flash freezing in liquid nitrogen. Data collection and structure solution was completed using the apo EN structure 1VYB for molecular replacement followed by refinement and manual building as described above.

## Plasmid nicking EN activity assay

The plasmid nicking assay was performed based on a previous assay[31] with the following modifications: the substrate was a supercoiled pUC57 plasmid containing the *E. coli* codon optimized L1 ORF0 sequence produced by miniprep (Qiagen). 8 nM EN WT and inhibitors or vehicle (10% DMSO) were incubated at room temperature for 1 h before adding 2 nM plasmid. Reactions were incubated at 37 °C for 3 h before stopping the reaction with heat inactivation (70 °C for 10 min) or addition of 50 mM EDTA. Reactions were run on a 1% agarose gel in 1X TAE buffer and visualized with ethidium bromide. Supercoiled plasmid without EN and linearized plasmid were included as controls.

## Fluorescent oligonucleotide activity assay and quantification

For the EN fluorescent oligonucleotide nicking activity assay, the hairpin sequence was adapted from a previous assay used for measuring APE1 activity[38]. The EN target sequence was added to the stem of the hairpin, the 6-FAM fluorescent tag was included at the 5′ end, and the DABCYL quencher was included at the 3′ end (5′-FAM-CGACTT TTAGATTGACACGCCATGTCGATCAATCTAAAAGTCG-DABCYL-3′). Reactions were completed in buffer containing 20 mM HEPES pH 7.5, 50 mM NaCl, 2.5 mM $MgCl_2$, and 10% DMSO. EN WT at 2.5 nM was incubated with inhibitors or vehicle for 1 h before adding 25 nM oligonucleotide. Fluorescence was measured at regular intervals at 37 °C with excitation 485 nm and emission 530 nm using a Synergy H1 plate reader with Gen5 software (BioTek). Initial rates for up to 10% turnover to product were normalized to no inhibitor control to calculate percent activity and $IC_{50}$ values were obtained using [inhibitor] vs. response−variable slope (four parameters) non-linear fit in GraphPad Prism for Windows. Curve fits were applied to each technical replicate containing one well for each inhibitor concentration to calculate three $IC_{50}$ values per independent experiment that were then averaged. The no inhibitor control and full inhibition by 50 mM EDTA were included in fit calculations to guide definition of the top and bottom of the curve fits[79].

The APE1 version of this assay was performed similarly to what was previously described[38] and detailed above for EN activity with the

following exceptions: substrate hairpin contained the Black Hole Quencher, initial rates were calculated for up to 20% turnover to product, APE1 (NEB, Cat # M0282S) at 0.002 U/μL was used, and buffer suitable for APE1 activity was used (10 mM HEPES pH 7.5, 100 mM NaCl, 100 mM KCl, 0.5 mM MgCl$_2$, 1% DMSO). The oligonucleotide sequence used for the APE1 assay can be found in Supplementary Data 1.

## Spectral shift by Monolith X

Spectral shift measurements[50] were completed using the Monolith X (NanoTemper). EN purified as described above was fluorescently labeled at lysine residues with the 2nd Generation Protein Labeling RED-NHS Kit (NanoTemper) following the kit instructions, but using the following labeling buffer: 10 mM HEPES pH 7.5, 150 mM NaCl, 0.1% pluronic F-127. Reactions containing 20 nM EN and inhibitors at indicated concentrations were incubated together for 1 h in buffer containing 10 mM HEPES pH 7.5, 150 mM NaCl, 0.1% Pluronic F-127, 5 mM EDTA, 10 mM MgCl$_2$, 1 mM DTT, and 5% DMSO. Samples were loaded into Monolith Premium Capillaries (NanoTemper) and fluorescence was measured with the Monolith X instrument using the MO. Control software. $K_d$ values were calculated in GraphPad Prism according to the law of mass action as described[50]. Capillaries were read three times to obtain three technical replicates, and each replicate containing one measurement at each inhibitor concentration was fit to calculate three $K_d$ values per independent experiment that were then averaged.

## Cell culture

HeLa Tet-On cells (Takara Bio Inc., Cat # 631183) were cultured at 37 °C in a 5% CO$_2$ supplemented air atmosphere. The cells were grown in DMEM with 10% FBS, 2 mM glutamine, and penicillin and streptomycin. Culture media was replaced every 2–3 days and when the cells were split. HeLa Tet-On cells were authenticated by Takara Bio using functional assays as described in the Certificate of Analysis and used at low passage in our laboratory. HeLa Tet-On cells containing plasmid pPM404[32], which expresses the L1 WT *ORFeus* sequence with a dual-luciferase reporter for use in the L1 retrotransposition assay[51], were a gift from the laboratory of Jef Boeke. HeLa Tet-On cultures containing pPM404 were maintained in 1 μg/mL puromycin to select for plasmid retention. HeLa Tet-On cells containing pPM404 were not further authenticated upon receipt from the laboratory of Jef Boeke and were used at low passage in our laboratory. LF1 human diploid fibroblasts cells were derived from embryonic lung tissue[57] and have been used in the laboratory since their isolation. LF1 cells were grown as previously described[16] under physiological oxygen conditions (92.5% N$_2$, 5% CO$_2$, 2.5% O$_2$) in Ham's F-10 nutrient mixture (Thermo Fisher) with 15% FBS, 2 mM glutamine, and penicillin and streptomycin. LF1 fibroblasts were authenticated as free from contamination by ATCC STR Profiling Service in 2019. None of the cell lines used is listed in the International Cell Line Authentication Committee (ICLAC) database as a commonly misidentified line.

For the DNA damage assays, plasmids expressing L1 FL WT (pDA007), FL EN- H230A (pDA025), FL EN-/RT- H230A/D702Y (pDA027), FL RT- D702Y (pAD034), EN WT, and EN- H230A *ORFeus* constructs were introduced into HeLa Tet-On cells by transfection with FuGENE HD (Promega) for 24 h. After removal of FuGENE HD, cells were selected with 1 μg/mL puromycin for 2 weeks before freezing. Cultures were tested regularly with MycoAlert Mycoplasma Detection Kit (Lonza).

## HeLa dual-luciferase L1 retrotransposition assay

The L1 retrotransposition assay was performed using HeLa Tet-On cells containing pPM404. Cells were maintained in 1 μg/mL puromycin and seeded into a 96-well plate at a density of 15,000 cells per well. Cells were induced with 1 μg/mL doxycycline and treated with inhibitors or vehicle (0.1% DMSO) as indicated for 48 h without replacing media. Cytotoxicity was evaluated with PrestoBlue Viability Reagent by incubating reagent with cells at 37 °C for 15 min and reading fluorescence with excitation 550 nm and emission 600 nm using a Cytation 5 Plate Reader with Gen5 software (BioTek). Luciferase activity luminescence was then measured with the Dual-Luciferase Reporter Assay System (Promega) also using the plate reader.

## Immunofluorescence staining and imaging

Cells for immunofluorescence were grown on coverslips in 24-well plates and washed with phosphate-buffered saline (PBS) prior to fixation in 4% paraformaldehyde for 20 min. Samples were treated with Permeabilization Buffer (PBS, 0.2% Triton) for 20 min and then Blocking Buffer (PBS, 0.02% Triton, 3% bovine serum albumin) for 20 min. Primary antibodies were diluted in Blocking Buffer as described below and incubated with samples for 2 h: human ORF1 monoclonal 4H1 mouse antibody (Millipore Sigma, 1:500), human ORF1 polyclonal rabbit antibody EPR22227-6 (Abcam, 1:500), γ-H2AX monoclonal mouse antibody JBW301 (Millipore Sigma, 1:1000), DNA/RNA hybrids monoclonal mouse antibody S9.6 (Kerafast, 1:200). Samples were washed with Blocking Buffer twice for 5 min each then treated with appropriate secondary antibody (Thermo Fisher) diluted 1:200 for 2 h: Alexa Fluor 488 goat anti-rabbit for ORF1 rabbit antibody, Alexa Fluor 546 donkey anti-mouse for DNA/RNA hybrids mouse antibody, or Alexa Fluor 647 donkey anti-mouse for ORF1 mouse antibody and γ-H2AX mouse antibody. Samples were washed twice with PBS for 5 min then treated with DAPI (1 μg/mL) for 15–30 min. Finally, coverslips were mounted onto slides with ProLong Gold Antifade Mountant (Invitrogen). Images were acquired with a Nikon Ti2-E Fluorescence Microscope using Nikon NIS-Elements software. Where applicable, images were scaled equivalently within each experiment in Adobe Photoshop to improve visualization in figures.

## γ-H2AX assay

HeLa Tet-On cells containing *ORFeus* plasmids as indicated above were induced with 2 μg/mL doxycycline and treated with inhibitors or vehicle for 24 h without media replacement before fixation as described above. Treatment with 40 μM etoposide for 24 h was used as a positive control for DNA damage. Images were acquired with a Nikon Ti2-E Fluorescence Microscope. Quantification of γ-H2AX was completed with CellProfiler and statistical analysis completed with GraphPad Prism for Windows (ROUT outlier correction with Q = 0.1% and one-way ANOVA followed by Dunnett's multiple comparisons test). An existing pipeline was modified to identify nuclei, detect γ-H2AX signal, and measure the mean γ-H2AX intensity for each nucleus with γ-H2AX signal. The pipeline efficacy was confirmed by manual validation of automatic nucleus detection. Images for each independent experiment were acquired during the same imaging session and with the same exposure.

## Neutral comet assay

HeLa Tet-On cells containing *ORFeus* EN WT or EN mutant plasmids were induced with 2 μg/mL doxycycline and treated with inhibitors or vehicle for 24 h without media replacement. As a control for DNA damage detection, cells were treated with 100 μM hydrogen peroxide for 20 min at 4 °C. The neutral comet assay was performed according to the Trevigen CometAssay Kit instructions with the following modification: samples were treated with Comet Assay Lysis Buffer (2.5 M NaCl, 100 mM EDTA, 10 mM Tris, and 0.1% Triton X-100, at pH 10) at 4 °C overnight. Cells were imaged with a Nikon Ti2-E Fluorescence Microscope using Nikon NIS-Elements software. Comet tail measurements were completed using the OpenComet plugin for ImageJ[56] and statistical analysis completed with GraphPad Prism for Windows (one-way ANOVA followed by Dunnett's multiple comparisons test). OpenComet efficacy was confirmed by manual curation of comet detection. Images for each independent experiment were acquired during the same imaging session and with the same exposure.

## Senescent cell culture and RT-qPCR

Replicatively senescent human diploid fibroblast LF1 cultures[57] were generated as previously described[16]. Briefly, cultures were split twice a week until the cells reached replicative exhaustion. Timing of senescence entry was designated as described[16]. Subsequently culture medium was replaced twice a week. After 1 month of senescence, cells were replated 1:1 into new 10 cm plates. Presence of senescence phenotypes was verified using several metrics as previously described[16]: cell enlargement, the senescence-associated β-galactosidase assay[58], RT-qPCR of SASP markers, and immunofluorescence detection of γ-H2AX foci, L1 ORF1 protein, and DNA/RNA hybrids. At or after 3 months of senescence, which corresponds to derepression of L1[16,80], cells were treated with inhibitors or no inhibitor control for 1 month with media and inhibitor replacement twice per week. Cultures were then harvested for RNA using Trizol (Invitrogen), followed by RNeasy Min Elute Cleanup Kit (Qiagen), and cDNA was generated using the TaqMan reverse transcription kit (Applied Biosystems). RT-qPCR was performed using the ViiA 7 Real-Time PCR System (Applied Biosystems) or QuantStudio 6 Pro Real-Time PCR System (Applied Biosystems). Primer sequences used were previously described[16] and can be found in Supplementary Data 1.

## RNA-seq transcriptomic analysis

RNA was extracted using Trizol (Invitrogen) followed by RNeasy MinElute Cleanup Kit (Qiagen). Sequencing data were generated by Azenta/GENEWIZ Inc. using the Illumina ultra-low input RNA kit and 2x150bp paired-end Illumina sequencing. Reads were preprocessed with fastp[81], aligned to the GRCh38.p14 human genome assembly with STAR[82], and assigned to genes using featureCounts in Subread[83]. Counts were normalized and differential gene expression analysis was performed by DESeq2[84], and genes with count values of 0 for all samples were filtered prior to subsequent analysis. Gene Set Enrichment Analysis[85] was performed for the interferon response[16] and the aging upregulated[59] gene sets added to the KEGG pathways, and the nominal $p$ values adjusted for false discovery rate by the Benjamini-Hochberg method[86]. Heat maps of normalized counts for each gene set were generated using Morpheus (https://software.broadinstitute.org/morpheus). The aging upregulated gene set can be found in the Molecular Signatures Database: https://www.gsea-msigdb.org/gsea/msigdb/human/geneset/DEMAGALHAES_AGING_UP.html.

## Reporting summary

Further information on research design is available in the Nature Portfolio Reporting Summary linked to this article.

# Data availability

The x-ray crystallography data generated and protein structures solved in this study have been deposited in the Protein Data Bank under the following accession codes: 8SP5 (LINE-1 retrotransposon endonuclease domain complex with Mn2+) and 8SP7 (LINE-1 retrotransposon endonuclease domain complex with tranexamic acid). The RNA-seq data generated in this study have been deposited in the Gene Expression Omnibus (GEO) database under accession code GSE244265. All other data supporting the findings of this study are available within the paper and its Supplementary Information. Source data are provided with this paper.

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

## Acknowledgements

This research was supported by NIH grants R01 AG016694 (J.M.S.), P01 AG051449 (J.M.S.), and T32 GM007601 (A.M.D. as trainee). Brown University has filed patents on the intellectual property described in this paper. This research used the AMX beamline of the National Synchrotron Light Source II, a U.S. Department of Energy (DOE) Office of Science User Facility operated for the DOE Office of Science by Brookhaven National Laboratory under Contract No. DE-SC0012704. The Center for BioMolecular Structure (CBMS) is primarily supported by the NIH (NIGMS) through a Center Core P30 Grant (P30GM133893), and by the DOE Office of Biological and Environmental Research (KP1607011). The authors would like to thank all members of the laboratories of John Sedivy, Gerwald Jogl, and Jill Kreiling for feedback and support. A.M.D. would specifically like to thank Bianca Kun, Anna Petrashen, and Radha Kalekar for assistance with senescent cell culture maintenance, as well as Maxfield Kelsey for guidance with RNA-seq analysis. The authors would like to thank Paolo Mita and Jef Boeke for providing the HeLa Tet-On pPM404 cell line. The authors would also like to thank the following core facility managers: Christoph Schorl (Genomics Core Facility), Mandar Naik (Structural Biology and Proteomics Core Facilities), and Geoff Williams (Leduc Bioimaging Facility).

## Author contributions

This study was conceptualized by all authors. J.M.S and G.J. jointly supervised all experiments. A.M.D. performed all experiments. A.M.D. drafted, revised, and edited the manuscript. J.M.S. and G.J. revised and edited the manuscript.

## Competing interests

J.M.S. is a cofounder and SAB chair of Transposon Therapeutics, holds equity in PrimeFour and Atropos Therapeutics, and consults for Atropos Therapeutics and Longaevus Technologies. Brown University has filed a pending patent (US20230139684A1) on the inhibitors described in this publication with J.M.S., G.J., and A.M.D. listed as inventors.
