## [Peer Review File · Nature Communications]

Identification and characterization of small molecule inhibitors of the LINE-1 retrotransposon endonucleaseREVIEWER COMMENTS

Reviewer #1 (Remarks to the Author):

This manuscript describes the identification and efficacy of new inhibitors targeting LINE-1-encoded endonuclease. Development of new pharmacological approaches that target transposable elements is important, as activation of these elements occurs in human cancers, neurodegenerative diseases, aging and other age-related disorders. D'Ordine and colleagues find that the newly developed inhibitors suppress LINE-1 associated DNA damage and senescence. This is important work that could be strengthened by addressing the following:

Major issues:

1. It appears that cells from three different experiments were pooled to generate the graphs in Fig. 5. The authors should consult with a statistician to determine if this is an appropriate way to analyze these data. It seems that the n is artificially high since each cell is counted as an individual n. Are these data still significant if the average fluorescence from each of three experiments is presented as a single datapoint (i.e. n=3 rather than n=56-752)? The wide range in the number of cells quantified per condition also seems like it may be a statistical issue.
2. It is somewhat disappointing that none of the newly identified compounds seem to outcompete 3TC. While I don't think this experiment should be required for acceptance, it would potentially increase the impact of the manuscript to determine if AD12 (or another top compound) has synergistic effects with 3TC.
3. The manuscript would be more accessible to a general audience if acronyms and assays were defined and described in more detail. APE-1, for example, is never defined as an endonuclease involved in base excision repair. Apo/holo structures are not defined, etc.
4. The transition to investigating effects of the endonuclease inhibitors on senescence is abrupt. The authors should provide the reader with more background on senescence, what is known about LINE-1-induced senescence, and why this is important.
5. It is unclear why the data in Fig. 4A are separated into four different graphs. It seems that the data could be normalized to the 0um treatment and presented on a single graph.
6. Fig. 6B: AD3 and AD14 are missing from CCL2 and IFN α . Why is this? The authors should also state why there are different treatment periods for the different targets (i.e. 5-6 months for IL6 but 3-4 months for MMP3).

Minor issues:

1. It is unclear why different shades of gray were used for each condition in Fig. 5. Is the shading meaningful?
2. Use of less acronyms throughout would help with readability.
3. Frequency of media/drug replacement on cells should be noted in the methods.

Reviewer #2 (Remarks to the Author):

This manuscript titled " Identification and characterization of small molecule inhibitors of the LINE-1 retrotransposon endonuclease" used a structure-based docking method to find EN inhibitors and provides Fluorescent oligonucleotide biochemical assay and Biophysical characterization to check its inhibition efficiency and specificity in vitro. Then, L1-induced DNA damage, L1 retrotransposition in cell culture, and inflammatory-related assay are used to check the effect of EN inhibitors in vivo. Even though some results are inconsistent between in vitro and in vivo studies, the authors provide the possibility in the discussion sections. The EN inhibitors identified in this study are the first inhibitors with novelty. The results of biochemical and cell-based experiments are clear, precise, and of potential

interest to the audience of Nature Communications, but the quality of this manuscript can be improved. Points to consider are:

1. The author uses structure-based docking to screen the inhibitors, which is the article's starting point and is essential. The author should provide more detailed information about the docking experiments. For example, are the docking sites of EN only at the active site or all surfaces of EN? On the other hand, does the docking model include the metal ion or not? These settings do affect the result of docking. If the docking site is the active site, the author should show the active site residues and surface for the understanding of readers. Figure S1 is unclear and does not indicate which region is the active site.
2. Before docking, the author used computational strategies to generate four groups of candidate compounds from multiple existing or filtered libraries. The detail of using computational methods to find the four groups of candidate compounds is lacking. The AD2 analogs group is clear, but how to find the other three groups is unclear, even though the author cites references 32 and 34 and provides some information in the Methods section. The author should also provide the number of candidate compounds for docking experiments.
3. The author indicates the active site of EN is similar to APE1. Because the APE1 inhibitors are the important candidate compounds, the superposition of the active sites of APE1 and EN becomes essential. The structural alignment (overall structure and active site) of APE1 and EN should be included in this manuscript. In addition, the author shows that AD1 is a promising inhibitor of APE1, but AD2, 3, 7, 9, 12, 32, 36, and 43 are not (Figure S5). In contrast, only AD1 is not a good inhibitor of EN, and others are good inhibitors. If their active sites are similar, why have these inhibitors shown the specificity? The author should explain this phenomenon from the point of view of structural comparison.
4. In Table S1, the value of R_{sym} in the last shell of two datasets must be wrong. The value should be lower than 1. The value of I/σ in the last shell of the L1 EN complex with Mn^{2+} is also unacceptable, and 0.6 is too low. Currently, the CC1/2 should be included in Table S1, which is more meaningful than R_{sym} or R_{merge} (See Kay's paper: Science 2012). In addition, there are 0.43% Ramachandran outliers in the structure of the L1 EN complex with tranexamic acid. The author should address the outlier residues or provide a reason.
5. The standard deviation is essential in the measurements of IC_{50} and K_d . The K_d in Figure 3 lacks the standard deviation (AD2 is $79.7 \pm xx \mu M$). On the other hand, the IC_{50} in Table 1 is measured by fluorescent oligonucleotide EN activity assay, similar to Figure 2. Why are the same experiment values different in Figure 2 and Table 1? Similarly, the K_d is measured by Spectral shift binding affinity in both Figure 3 and Table 1. Why the K_d values are also different by measuring the same experiment?
6. In Figure S3, the author should label the bands of linear, open circular, and supercoiled plasmid. In the AD1 figure, the related positions of uncut plasmid (U) and linearized plasmid (L) are the same, which is unreasonable.
7. In Figure S8. The map is too complicated. The author can only show the omitted maps of the metal ions with coordination waters and ligands (tranexamic acid). The omitted maps should be the $F_o - F_c$ map, not the $2F_o - F_c$ map.
8. The tranexamic acid in the EN/fragment (tranexamic acid) complex structure (PDB ID: 8SP7) determines fragment location for docking analogs. Can tranexamic acid also inhibit EN or have a strong interaction with EN? The author should explain why tranexamic acid can be used to determine fragment location. Why compounds with this fragment or similar fragments could be candidates for EN?

Overall, this study starts from the protein structure but lacks the analysis and discussion about the structures. The structure analysis and comparison in the discussion section will help explain the lack of consistency or unexpected results.

Reviewer #3 (Remarks to the Author):

Within the human genome, the only known autonomous retrotransposon is the LINE1 or L1 element. Insertion of the element within specific genes has been associated with a number of diseases. Beyond genomic insertions, L1 elements have been associated with negative cellular outcomes resulting from DNA damage and cellular stress that impact tumor progression. D'Ordine et al. in this manuscript focus on the identification and characterization of small molecule inhibitors targeting the endonuclease function of ORF2. This endonuclease (EN) is structurally and to some extent functionally similar to the base excision repair enzyme, apurinic/apyrimidinic endonuclease 1 (APE1), the primary difference being that APE1 is an incredibly fast enzyme while EN is by comparison quite slow. The L1 EN is a very interesting target for development of chemical tools with potential mechanistic and therapeutic uses. Initial identification of small molecules involved use of multiple docking programs and several different libraries and was facilitated by determination of Mn²⁺ and a tranexamic fragment EN structures. Overall, the workflow for the studies is sound and findings are of general interest. However, it is very difficult to follow the logic in characterizing inhibitors with near mM IC₅₀ values for inhibition of EN activity in follow-up assays. These compounds are not very promising in terms of targeting EN. The contention that these compounds would inhibit retrotransposition through direct targeting of EN in cells at concentrations well below the IC₅₀ values is implausible. A more careful consideration of off target effects in the various assays would be appropriate.

Specific comments.

1. Introduction. In the following sentence, the type of DNA damage should be explicitly stated. "Additionally, the DNA damage induced by L1 activity is driven in part by the EN." In general, single strand breaks are readily repaired. Double strand breaks are potentially more cytotoxic.
2. Introduction. Rationale for pursuing L1 EN inhibitors could use further detail. "Finally, development of EN inhibitors would enable combining pharmacological inhibition of both catalytic L1 domains for potential synergistic effects." Clearly, EN inhibitors will not be useful for correcting problems that arise from insertion of L1 elements within genes that give rise to disease.
3. Note. The base excision repair enzyme used for comparison in this study is APE1, not APE-1.
4. Results. Clarify the screening procedure. Screening was done using the Mn²⁺ bound structure? Was the metal used in a fully coordinated state? Or stripped of coordinating waters? Was consensus scoring to evaluate hits?
5. The docking results for all of the compounds tested in this study should be shown in figures with appropriately labeled residues if not in the main text then in the Supporting Information.
6. Although many of the active site residues in APE1 are also found in EN, there is one notable difference. The residue equivalent to S202 in EN is W280 in APE1; the presence of W280 impacts the size and shape of the pocket. Some discussion regarding similarities and differences in the active sites of these enzymes should be included.
7. None of the APE1 endonuclease inhibitors screened for EN inhibition have been shown to directly bind APE1. This point should be made clear in the manuscript.
8. Compounds were not screened for DNA-binding activity? This is an important control when the in vitro assay includes a DNA substrate. The compounds were cross screened for APE1 endonuclease inhibition. With the exception of AD1, which looks like a DNA intercalator, the compounds showed modest inhibition at 25 μ M, the most potent being AD17. Why were the cmpds screened at 25 μ M when many of them have IC₅₀ values for inhibition of EN that are much higher than 25 μ M?
9. IC₅₀ values in representative plots shown in Fig. 2 do not match the values in Table 1. How many parameters were used for the non-linear fit of the data in Fig. 2? How were replicate data combined to obtain the values in Table 1? Three sets of three replicates were averaged or all 9 sets of data were subjected to a non-linear fit? The standard deviations for the IC₅₀ values and K_d measurements are

very large. In general, these values would be maximally 10-15%. For many of the compounds the values are much larger than that. Same comments apply to the K_d data shown in Fig. 3.

10. Table 1: How were the concentrations of compound selected for the RetroT efficiencies? The values appear to be a little random and not related to the IC_{50} values in some cases. For example, AD7 has an IC_{50} value of 0.875 mM but 20 μ M reduced RetroT efficiency to 42%. It's very difficult to argue that with an IC_{50} value of 0.875 mM, treatment with 20 μ M AD7 is on target. The same could be said for AD12. This is very unlikely to be on target.

11. Discussion. It is not possible for good cellular permeability to account for differences in target engagement seen for the compounds as suggested in the following statement. "On the other hand, several inhibitors showed relatively low efficacy in vitro but high activity in the retrotransposition assay: AD7, AD9, AD12, AD32, AD36, and AD43. One possibility is that these compounds have higher cell permeability, allowing them to achieve higher intracellular concentrations." How exactly would this happen?

12. Table S1. Include $CC_{1/2}$ values for each data set. R_{sym} and R_{merge} are not the same. Which value is presented in the Table? Also, the R_{sym} or R_{merge} for the highest resolution shell of 6.15 and $I/\sigma I$ of 0.6 for the Mn^{2+} structure suggests that the data are not usable to the resolution reported. The data should be cut back to lower resolution.

13. Fig. S1, panel A. Showing a surface rendering with no residues labeled or highlighted in different colors is not particularly informative. Either color code residues of interest in the surface rendering or use a semi-transparent surface rendering and show residues of interest highlighted in color as stick renderings.

14. Coordinating ligands for the Mn^{2+} ion should be indicated through dashed lines and bond distances provided in Fig. S1 or in the text.

15. What interactions are involved in the binding of tranexamic acid to EN? No hydrogen-bonding interactions are indicated in Figure S1. Fo-Fc density prior to fitting of the fragment should be shown to support the fitting of the fragment.

16. The spectral shift binding assay is not standardly used to validate direct binding of small molecules to proteins.

17. Figures. The figure caption for Figure S2 does not include sufficient information. The text indicates that these compounds are APE1 inhibitors reported in the literature, and yet the compounds are renamed here as AD1-AD15? Then a subset of these compounds are shown docked in EN. Full inhibition of AD3 at 1 mM does not seem like a meaningful result. Many compounds would inhibit at this concentration.

Response to Review

We sincerely thank the reviewers for their time and effort to provide feedback on our initial submission. We greatly appreciate their interest and constructive criticism of our work, and we were able to address the great majority of their points. To summarize the major issues:

- We performed additional experiments, in particular, we ran replicates of the biochemical and biophysical characterization of several inhibitors to confirm their efficacy. We also performed control experiments to address non-specific binding of inhibitors to DNA.
- We reanalyzed and changed the presentation of our DNA damage experiments, addressed statistical issues, and clarified the treatment of replicate experiments.
- We addressed questions regarding the statistics of the crystal structures and we generated improved figures of these structures.
- We added more detail regarding the molecular docking approaches used to evaluate candidate compounds, and revised and expanded the associated figures.
- Two reviewers felt that a structural comparison of the L1 EN and the related cellular enzyme APE1 was important for understanding the differences in inhibitor efficacy for these two targets. We thus included a discussion of these differences in the text and created a figure to show them visually.

In the text below, the comments of the reviews appear in black font, and our response to each point is shown in blue font. We provide line numbers for all the major changes and/or additions we made to the revised manuscript file. In the manuscript file as well as the supplement file substantive changes are highlighted in red font. We also made numerous small changes in sentence structure and word choice throughout to improve clarity and readability; to avoid clutter these are not highlighted.

Reviewer 1

This manuscript describes the identification and efficacy of new inhibitors targeting LINE-1-encoded endonuclease. Development of new pharmacological approaches that target transposable elements is important, as activation of these elements occurs in human cancers, neurodegenerative diseases, aging and other age-related disorders. D'Ordine and colleagues find that the newly developed inhibitors suppress LINE-1 associated DNA damage and senescence. This is important work that could be strengthened by addressing the following:

Major issues:

1. It appears that cells from three different experiments were pooled to generate the graphs in Fig. 5. The authors should consult with a statistician to determine if this is an appropriate way to analyze these data. It seems that the n is artificially high since each cell is counted as an individual n . Are these data still significant if the average fluorescence from each of three experiments is presented as a single datapoint (i.e. $n=3$ rather than $n=56-752$)? The wide range in the number of cells quantified per condition also seems like it may be a statistical issue.

We appreciate the suggestion that we revise the presentation and mathematical treatment of this series of experiments. We consulted with a statistician and after reviewing all the data we made the following changes:

- We agree that nuclei from independent experiments should not be pooled. At the recommendation of the statistician, we now show one representative independent experiment in Fig. 5. The general approach for presenting these DNA damage assays is now based on a similar experiment also measuring L1-induced γ -H2AX nuclear signals

(Ardeljan et al. (2020) Nat. Struct. Mol. Biol. 27: 168-178). As they have done (and also recommended by our statistician) we continue to score each nucleus as a discrete data point, to capture the biological variability within the experiment. Furthermore, we have presented the entire dataset (all data from all independent experiments) in a new table (Table S4) so that the reader can inspect the reproducibility from one independent experiment to another. This table also shows the means and standard deviations across all independent experiments. Finally, we have included in the Legend to Fig. 5 the total number of nuclei analyzed for each treatment (lines 1013-1022).

- To improve our confidence in the average fluorescence signal for each nucleus, we reanalyzed our images with a less stringent cutoff for detectable γ -H2AX signal. This increased the dynamic range and therefore the accuracy of the measurements. All values scored remain above the background threshold.
- Review of the comet assay tail moment data showed lower tail moments for some inhibitors when compared to the no dox and/or EN- control, while the tail length values remained above the controls. As a result, we have decided to show the tail length data only, which we believe to be a more accurate measurement of the DNA damage in this assay based on our controls.

2. It is somewhat disappointing that none of the newly identified compounds seem to outcompete 3TC. While I don't think this experiment should be required for acceptance, it would potentially increase the impact of the manuscript to determine if AD12 (or another top compound) has synergistic effects with 3TC.

We agree with the reviewer that combined treatments with RT and EN inhibitors would be interesting to test, and we are working on this for future publications. This would be particularly interesting for senescent cells where effects of individual compounds are modest. Unfortunately, late senescent cells take over 4 months to prepare. For this paper we prioritized screening a larger number of candidate compounds rather than testing multiple combinations of inhibitors. We acknowledge this point in the Introduction (lines 114-117) and Discussion (lines 426-428).

3. The manuscript would be more accessible to a general audience if acronyms and assays were defined and described in more detail. APE-1, for example, is never defined as an endonuclease involved in base excision repair. Apo/holo structures are not defined, etc.

We apologize for this. We have now defined APE1 as a base excision repair enzyme in the Introduction (line 95-96) and clarified when apo or complex proteins are described (for example lines 471-473, 540). We have also added all acronym definitions at point of first use.

4. The transition to investigating effects of the endonuclease inhibitors on senescence is abrupt. The authors should provide the reader with more background on senescence, what is known about LINE-1-induced senescence, and why this is important.

This is a helpful suggestion for which we are grateful. We have now provided more background on senescence and its connection to L1 in the Introduction (lines 60-65), as well as in the Results section (lines 299-302). We have additionally clarified in these comments that L1 does not induce senescence, but rather that L1 expression increases during late senescence and appears to reinforce existing inflammation.

5. It is unclear why the data in Fig. 4A are separated into four different graphs. It seems that the data could be normalized to the 0um treatment and presented on a single graph.

We agree that this is a potential source of confusion. However, the data are presented in separate graphs because they represent independent assay runs done on separate occasions with different batches of cells. Each run was a separate experiment with its own 0 μ M data point. We believe that presenting the data in this way is valuable because it preserves experiment to experiment variability. We have however carefully clarified in the Figure Legend that these are independent experiments (lines 995-996), that average retrotransposition frequencies (of at least triplicate independent experiments) are presented in Table 1 (lines 271-272), and that the full data for all replicates are found in Table S3 (lines 1058-1059).

6. Fig. 6B: AD3 and AD14 are missing from CCL2 and IFN α . Why is this? The authors should also state why there are different treatment periods for the different targets (i.e. 5-6 months for IL6 but 3-4 months for MMP3).

We have updated the Figure Legend (lines 1033-1036, 1039-1040) to clarify that these results are from independent cultures of senescent cells in which the cells were treated for 4 weeks, but the treatments were started at different times after entry of the culture into senescence. This was done simply for experimental convenience, and was based on previous work from our laboratory (De Cecco et al. (2013) *Aging Cell* 12: 247-256; De Cecco et al. (2019) *Nature* 566: 73-78), which showed that L1 expression begins around 3 months into senescence (which we termed “late senescence”) and remains relatively constant thereafter (as does general gene expression, such as expression of the interferon pathway, which defines late senescence). This means that beginning treatments at 3, 4, or 5 months into senescence yields similar results. Since late senescent cells are very time consuming to prepare, we generally like to use any late senescent cells we have on hand. We have added these details to the Methods (line 662-664) and Results (lines 305-307) sections. Additionally, AD3 and AD14 were not present for CCL2 and IFN α because this was the first trial chronologically speaking, at which point we had limited cells and inhibitor available, so we prioritized testing AD12 as it had the strongest effect in the retrotransposition assay. To clarify this issue for the readers, we have also adjusted the arrangement of the panels to reflect the order in which the experiments were done.

Minor issues:

1. It is unclear why different shades of gray were used for each condition in Fig. 5. Is the shading meaningful?

We apologize for the confusion regarding the shading. We had made the individual points partially transparent to improve visualization of multiple overlapping data points, which however made some areas appear darker due to the presence of more nuclei. We have now removed this effect.

2. Use of less acronyms throughout would help with readability.

We have removed some acronyms (such as SVA and BSA) and defined all the remaining acronyms (for example, line 88).

3. Frequency of media/drug replacement on cells should be noted in the methods.

We have added these details to the Methods in the sections pertaining to cell culture maintenance and the retrotransposition, DNA damage, and senescence assays (lines 591-592, 605, 629, 642, 657-658, 663-664).

Reviewer 2

This manuscript titled " Identification and characterization of small molecule inhibitors of the LINE-1 retrotransposon endonuclease" used a structure-based docking method to find EN inhibitors and provides Fluorescent oligonucleotide biochemical assay and Biophysical characterization to check its inhibition efficiency and specificity in vitro. Then, L1-induced DNA damage, L1 retrotransposition in cell culture, and inflammatory-related assay are used to check the effect of EN inhibitors in vivo. Even though some results are inconsistent between in vitro and in vivo studies, the authors provide the possibility in the discussion sections. The EN inhibitors identified in this study are the first inhibitors with novelty. The results of biochemical and cell-based experiments are clear, precise, and of potential interest to the audience of Nature Communications, but the quality of this manuscript can be improved. Points to consider are:

1. The author uses structure-based docking to screen the inhibitors, which is the article's starting point and is essential. The author should provide more detailed information about the docking experiments. For example, are the docking sites of EN only at the active site or all surfaces of EN? On the other hand, does the docking model include the metal ion or not? These settings do affect the result of docking. If the docking site is the active site, the author should show the active site residues and surface for the understanding of readers. Figure S1 is unclear and does not indicate which region is the active site.

These are useful suggestions which we have incorporated. We now state in the Methods (lines 473-476) and Results (lines 162-164) sections that the docking site used was the DNA binding surface containing the active site residues and modified Fig. S1A and Fig. S2 to clearly indicate this area. We have also clarified in the Methods (lines 471-473), Results (lines 188-189), and Fig. 1 Legend (lines 921-922) that the apo EN structure was used for initial docking of APE1 inhibitors and AD2 analogs, and then once we solved the structure with Mn^{2+} bound, we used this structure for evaluating all other compounds.

2. Before docking, the author used computational strategies to generate four groups of candidate compounds from multiple existing or filtered libraries. The detail of using computational methods to find the four groups of candidate compounds is lacking. The AD2 analogs group is clear, but how to find the other three groups is unclear, even though the author cites references 32 and 34 and provides some information in the Methods section. The author should also provide the number of candidate compounds for docking experiments.

We agree that adding this information improves the paper. Accordingly, we have provided additional details on the workflow for the ZINC subsets, pharmacophore, and fragment analogs docking groups in both the Methods (lines 479-485, 489-497) and Results (lines 192-194, 196-202) sections. We have also included the total number of compounds docked for each group in the Methods (lines 478, 482, 491-492, 497) and listed the inhibitors from each group in the Results (lines 181-182, 185-187, 195-196, 205-206). It is important to note that the databases we used to obtain files for docking are frequently updated. Hence, the exact number of compounds obtained by following our workflow, as detailed in the Methods, will likely vary slightly depending on when the search is performed, and we have added a note concerning this in the Methods (lines 498-499).

3. The author indicates the active site of EN is similar to APE1. Because the APE1 inhibitors are the important candidate compounds, the superposition of the active sites of APE1 and EN becomes essential. The structural alignment (overall structure and active site) of APE1 and EN

should be included in this manuscript. In addition, the author shows that AD1 is a promising inhibitor of APE1, but AD2, 3, 7, 9, 12, 32, 36, and 43 are not (Figure S5). In contrast, only AD1 is not a good inhibitor of EN, and others are good inhibitors. If their active sites are similar, why have these inhibitors shown the specificity? The author should explain this phenomenon from the point of view of structural comparison.

Thank you for this useful suggestion. We have added the EN and APE1 superimposition as Fig. S2. We have added discussion of the differences between the active sites, as also suggested by Reviewer 3, when the APE1 inhibitors are introduced as candidate compounds in Results (lines 157-162), as well as when the results of the APE1 activity assay are described in Results (lines 234-237). This in combination with further delineation of the areas included in docking calculations in Fig. S1 and Fig. S2 should help improve the reader's understanding of how APE1 inhibitors presented a good starting point for EN inhibitor testing in the absence of any known inhibitors. Furthermore, this discussion explains how cross-reactivity can be avoided due to the differences in binding of DNA downstream of the scissile bond.

4. In Table S1, the value of R_{sym} in the last shell of two datasets must be wrong. The value should be lower than 1. The value of I/σ in the last shell of the L1 EN complex with Mn²⁺ is also unacceptable, and 0.6 is too low. Currently, the CC1/2 should be included in Table S1, which is more meaningful than R_{sym} or R_{merge} (See Kay's paper: Science 2012). In addition, there are 0.43% Ramachandran outliers in the structure of the L1 EN complex with tranexamic acid. The author should address the outlier residues or provide a reason.

We apologize for this confusion. We have added the CC1/2 values as requested (both by Reviewer 2 and Reviewer 3). We had originally set the resolution limit of the EN complex with Mn²⁺ as 1.92Å, which resulted in a CC1/2 of 0.350 in the inner shell, based on the CC1/2 > 0.3 guideline (Karplus & Diederichs (2015) Curr. Opin. Struct. Biol. 34: 60-68). We have now adjusted the resolution to 2.23 Å in order to improve the I/σ in the last shell and R_{merge} statistics. This resulted in very minimal changes to the structure (RMSD=0.048). For the EN complex with tranexamic acid, the outlier is a proline present in the flexible hairpin loop. There is density present for this residue in the omit map we generated to help us solve the structure. Also, this residue was consistently the only outlier shown after iterative rounds of refinement, even after the residue was corrected in the model used for refinement. Therefore, we have not updated this structure. See Table S1 for updated statistics for both structures.

5. The standard deviation is essential in the measurements of IC₅₀ and K_d. The K_d in Figure 3 lacks the standard deviation (AD2 is 79.7 ± xx μM). On the other hand, the IC₅₀ in Table 1 is measured by fluorescent oligonucleotide EN activity assay, similar to Figure 2. Why are the same experiment values different in Figure 2 and Table 1? Similarly, the K_d is measured by Spectral shift binding affinity in both Figure 3 and Table 1. Why the K_d values are also different by measuring the same experiment?

Both Reviewer 2 and Reviewer 3 pointed out the confusion in reporting the IC₅₀ and K_d values in Fig. 2 and Fig. 3 and Table 1. To clarify, Fig. 2 and Fig. 3 show one independent assay (experiment) comprised of 3 technical replicates for which a mean IC₅₀ or K_d value was calculated, while Table 1 averages IC₅₀ and K_d values across multiple independent assays (experiments) and includes the standard deviations for each inhibitor. We have clarified this in the Results (lines 222-223, 253-255). We have also added Table S3 with the individual replicate values used to calculate the means shown in Table 1 (see discussion below, Reviewer 3 point 9). We believe this more accurately shows the variation among independent assays. We have clarified this and referred to either Table 1 or Table S3 as applicable in the Figure Legends of

Fig. 2, Fig. 3, and Table 1 (lines 953-954, 968, 1058-1059). Additionally, Fig. 2 and Fig. 3 both show the standard deviations of each measurement as error bars based on the 3 replicates within each assay.

6. In Figure S3, the author should label the bands of linear, open circular, and supercoiled plasmid. In the AD1 figure, the related positions of uncut plasmid (U) and linearized plasmid (L) are the same, which is unreasonable.

We have added the labels for the supercoiled, nicked, and linearized plasmid forms on all gels. We have also replaced the AD1 figure with another gel run on a different occasion, which more clearly shows the differences between the positions of the uncut (supercoiled), linearized, and nicked plasmids as seen in the gels for the other compounds.

7. In Figure S8. The map is too complicated. The author can only show the omitted maps of the metal ions with coordination waters and ligands (tranexamic acid). The omitted maps should be the Fo-Fc map, not the 2Fo-Fc map.

We agree with the reviewer and we have calculated the omit map for both complexes and displayed the Fo-Fc maps in Fig. S1. In order to calculate these maps, the ligands and coordinating waters were omitted. These maps show clear difference density corresponding to the waters, ion, and ligand.

8. The tranexamic acid in the EN/fragment (tranexamic acid) complex structure (PDB ID: 8SP7) determines fragment location for docking analogs. Can tranexamic acid also inhibit EN or have a strong interaction with EN? The author should explain why tranexamic acid can be used to determine fragment location. Why compounds with this fragment or similar fragments could be candidates for EN?

Tranexamic acid is a fragment included as part of the Jena Bioscience Frag Xtal Screen. Solving the structure with this fragment bound has allowed us to determine where it interacts with the EN active site when present at sufficiently high concentrations, and where compounds with similar moieties may also interact. We have added more discussion and justification of this approach in the Results section (lines 196-205), and more details on the protocol in Methods (lines 492-497). This compound is small (157 Da) and does not inhibit sufficiently to calculate an IC_{50} in the fluorescent oligonucleotide assay. We observed ~75% inhibition at the highest soluble concentration in the fluorescent oligonucleotide assay, 25 mM, which is lower than the concentration used in the soak used to solve the crystal structure of the complex, which was 50 mM (see lines 533-541 for crystallization details). We also did not detect binding by spectral shift, probably due to the fact that the highest concentration that we could test under the buffer conditions used for this instrument was 12.5 mM. We believe we did not detect weak binding due to the relatively small size of this molecule compared to our other inhibitors. This is supported by the observation that AD14 (185 Da), the smallest inhibitor we evaluated, has the lowest magnitude change in spectral shift ratio compared to the other inhibitors.

Overall, this study starts from the protein structure but lacks the analysis and discussion about the structures. The structure analysis and comparison in the discussion section will help explain the lack of consistency or unexpected results.

We have updated our structures and added more discussion of them based on feedback from Reviewers 2 and 3, as described in detail in the responses to each comment. Specifically, we have provided more detail regarding our docking procedures in the Methods section (lines 470-

501) and through updated supplemental figures (Fig. S1, S4). We have also included a comparison to APE1 to help explain inhibitor results in the Results section at several points (lines 157-162, 234-237) and through a new supplemental figure (Fig. S2). In the Discussion (lines 413-417), we mention how the open DNA binding surface of the EN could be suitable for a wide range of inhibitor scaffolds to minimize off-target effects on enzymes that share similarities with parts of this surface, such as APE1. We also present in the Discussion several possibilities why some inhibitors may interact differently with the EN in the context of the full-length ORF2 versus the isolated domain (see paragraph lines 376-380).

Reviewer 3

Within the human genome, the only known autonomous retrotransposon is the LINE1 or L1 element. Insertion of the element within specific genes has been associated with a number of diseases. Beyond genomic insertions, L1 elements have been associated with negative cellular outcomes resulting from DNA damage and cellular stress that impact tumor progression. D'Ordine et al. in this manuscript focus on the identification and characterization of small molecule inhibitors targeting the endonuclease function of ORF2. This endonuclease (EN) is structurally and to some extent functionally similar to the base excision repair enzyme, apurinic/apyrimidinic endonuclease 1 (APE1), the primary difference being that APE1 is an incredibly fast enzyme while EN is by comparison quite slow. The L1 EN is a very interesting target for development of chemical tools with potential mechanistic and therapeutic uses. Initial identification of small molecules involved use of multiple docking programs and several different libraries and was facilitated by determination of Mn²⁺ and a tranexamic fragment EN structures. Overall, the workflow for the studies is sound and findings are of general interest. However, it is very difficult to follow the logic in characterizing inhibitors with near mM IC₅₀ values for inhibition of EN activity in follow-up assays. These compounds are not very promising in terms of targeting EN. The contention that these compounds would inhibit retrotransposition through direct targeting of EN in cells at concentrations well below the IC₅₀ values is implausible. A more careful consideration of off target effects in the various assays would be appropriate. Specific comments.

1. Introduction. In the following sentence, the type of DNA damage should be explicitly stated. "Additionally, the DNA damage induced by L1 activity is driven in part by the EN." In general, single strand breaks are readily repaired. Double strand breaks are potentially more cytotoxic.

The reviewer is quite correct. However, empirically it has been shown by several groups that expression of active EN in cells results in DNA damage which is presumably dsDNA breaks (γ -H2AX foci). We have clarified in the manuscript that L1 expression has been implicated in making double strand DNA breaks in the sentence following the one indicated by the reviewer (Introduction, lines 99-102). A very recent paper (Thawani et al. (2023) Nature doi.org/10.1038/s41586-023-06933-5) provides some mechanistic insights into a possible explanation for how EN single-strand nicking can result in dsDNA breaks.

2. Introduction. Rationale for pursuing L1 EN inhibitors could use further detail. "Finally, development of EN inhibitors would enable combining pharmacological inhibition of both catalytic L1 domains for potential synergistic effects." Clearly, EN inhibitors will not be useful for correcting problems that arise from insertion of L1 elements within genes that give rise to disease.

We agree with the point of the reviewer. We have added more rationale for the EN and L1 in general as targets for inhibitors in the Introduction, such as the beneficial effects of RT inhibition

in aged mice and the potential for experiments testing the relative contributions of ORF2 domains to L1-induced phenotypes (lines 90-91, 96-98, 114-116). Our own research was stimulated by the ability to readily produce large amounts of EN *in vitro* for screening, which was not possible for RT when we started this project. We agree that these inhibitors would not be used to correct L1 insertions after they occur, but rather to reduce ongoing insertions, DNA damage, and more broadly the activity of the proteins encoded by L1. We have clarified this sentence, highlighted by the reviewer, to suggest that using RT and EN inhibitors together could have additive effects (lines 116-117), as Reviewer 1 also points out (see point 2 above).

3. Note. The base excision repair enzyme used for comparison in this study is APE1, not APE-1.

Thank you, we have changed the spelling to APE1.

4. Results. Clarify the screening procedure. Screening was done using the Mn²⁺ bound structure? Was the metal used in a fully coordinated state? Or stripped of coordinating waters? Was consensus scoring to evaluate hits?

We have added more detail to the docking Methods section (lines 470-501), as requested by Reviewer 2 as well (see points 1 and 2 above). Screening was done using the Mn²⁺ bound structure for all rounds other than evaluating the APE1 inhibitors and AD2 analogs. The coordinating waters were removed prior to docking (as well as all waters as part of the receptor preparation procedure), and we have added this detail to the Methods section (lines 476-477). We did not use consensus scoring incorporating multiple scoring function outputs from each program, but rather compounds were evaluated based on the default ranking metric from each program and the similarity of results across programs. We have now stated this in the Methods (lines 499-501) and Results (lines 178-181) sections.

5. The docking results for all of the compounds tested in this study should be shown in figures with appropriately labeled residues if not in the main text then in the Supporting Information.

As requested, we have included these docking results in Fig. S4. We have labeled the residues that form polar contacts with the compounds, as well as shown the surface view overlaying results from multiple programs where applicable.

6. Although many of the active site residues in APE1 are also found in EN, there is one notable difference. The residue equivalent to S202 in EN is W280 in APE1; the presence of W280 impacts the size and shape of the pocket. Some discussion regarding similarities and differences in the active sites of these enzymes should be included.

We thank Reviewers 2 and 3 for suggesting we further compare the EN and APE1. We have added Fig. S2 containing the superimposition of the proteins to illustrate the S202 and W280 comparison, and included references to this figure when APE1 is first introduced and when EN inhibitors are tested against APE1 in Results (lines 157-164, 234-237) (see Reviewer 2, point 3 above).

7. None of the APE1 endonuclease inhibitors screened for EN inhibition have been shown to directly bind APE1. This point should be made clear in the manuscript.

We have added this acknowledgment when introducing APE1 inhibitor AD2 used as a preliminary weak EN inhibitor in Results (lines 176-177). In this sentence we also mention that

direct binding of AD1 to APE1 has been shown by fluorescence measurements and by competition with another known APE1 inhibitor, E3330, which binds to the APE1 endonuclease active site.

8. Compounds were not screened for DNA-binding activity? This is an important control when the in vitro assay includes a DNA substrate. The compounds were cross screened for APE1 endonuclease inhibition. With the exception of AD1, which looks like a DNA intercalator, the compounds showed modest inhibition at 25 μ M, the most potent being AD17. Why were the compounds screened at 25 μ M when many of them have IC50 values for inhibition of EN that are much higher than 25 μ M?

We have performed the additional experiment suggested by the reviewer, namely, we have screened all our inhibitors for DNA binding using a spectral shift assay (Table S2) and referenced this experiment in the Results section (lines 250-253). We did the DNA binding checks of inhibitors at the highest concentration used for the EN binding spectral shift assay. In these experiments we used the oligonucleotide substrate labeled with Cy5 as the target. None of the inhibitors showed detectable binding with the exception of AD16; however the magnitude of the spectral shift response for AD16 was not as high as for our positive control, SYBR Gold. AD16 shows stronger EN binding relative to many other EN inhibitors (Fig. 3, Table 1), supporting that its predominant mechanism of inhibition is through EN binding rather than non-specific DNA binding. This general lack of DNA binding is consistent with the fact that we did not observe toxicity in our cell-based assays at the concentrations tested, as toxicity could occur as a result of non-specific DNA interaction. The absence of significant inhibitor/DNA binding is also supported by the fact that we did not see widespread inhibitor efficacy in the APE1 assay with the oligonucleotide substrate (which is the same length and has the same hairpin in both EN and APE1 assays). In the APE1 activity assay the compounds were tested at 25 μ M based on their solubility in the assay buffer (the EN can tolerate higher concentrations of DMSO than APE1). For compounds with higher solubilities, we did test them at 250 μ M and found them to not inhibit, while the weak APE1 inhibitor AD2 inhibited at this concentration. We also list the compounds that were tested at 250 μ M in the Results section (lines 231-233).

9. IC50 values in representative plots shown in Fig. 2 do not match the values in Table 1. How many parameters were used for the non-linear fit of the data in Fig. 2? How were replicate data combined to obtain the values in Table 1? Three sets of three replicates were averaged or all 9 sets of data were subjected to a non-linear fit? The standard deviations for the IC50 values and Kd measurements are very large. In general, these values would be maximally 10-15%. For many of the compounds the values are much larger than that. Same comments apply to the Kd data shown in Fig. 3.

We used the four-parameter [inhibitor] vs. response – variable slope non-linear fit in GraphPad Prism to calculate the IC50 values, and we have now included this extra detail in the Methods section (lines 564-565) and Figure Legend for Fig. 2 (lines 950-951).

Both Reviewer 2 and Reviewer 3 pointed out the confusion in reporting the IC50 and Kd values in Fig. 2 and Fig. 3 and Table 1. As indicated in our response to Reviewer 2 (point 5 above), Fig. 2 and Fig. 3 show one independent assay (experiment) comprised of 3 technical replicates for each concentration. These values were then used to calculate one IC50 or Kd value per independent assay (experiment) shown in Fig. 2 and Fig. 3, while Table 1 averages IC50 and Kd values across multiple independent assays (experiments) and includes the standard deviations for these averages. We believe this more accurately shows the variation among independent assays (discussed below). We have clarified this in the Figure Legends of both

figures (lines 953-954, 968). Additionally, Fig. 2 and Fig. 3 both show the standard deviation of each measurement as error bars using the 3 replicates within each assay. It can be seen here that the technical reproducibility in a single experiment is very good.

Concerning data in Table 1 (combined IC50 and Kd values) these were averaged from experiments run at different times, and in some cases used separate lots of inhibitors and/or different batches of purified EN. We believe this was a source of some variation. We also reviewed the existing replicates and excluded the results from a few independent assays. In all cases another replicate was completed to replace it and this resulted in lower standard deviations than previously. Our reasoning was as follows:

- AD13, IC50: one replicate was removed due to inability of GraphPad Prism to calculate a confidence interval for the best fit IC50 value.
- AD17, Kd: one replicate was removed due to capillary adsorption present in some technical replicates, which interferes with fluorescence measurements needed to generate the binding curve and contributes to increased noise.
- AD18, Kd: one replicate was removed due to a poor fit and larger confidence interval calculated by the Monolith X software relative to the other replicates.

We apologize for not catching these inconsistencies previously. All the data for all the replicates used to calculate the IC50, Kd, and retrotransposition assay values in Table 1 can be found in Table S3, which we also reference in the Table 1 Legend (lines 1058-1059). We believe the somewhat higher standard deviations among spectral shift measurements may be due to the use of several different batches of labeled EN across independent experiments.

Concerning the issue of the magnitude of standard deviations between independent experiments, we did a literature search to get a sense for standard practices used in the field. We have noted multiple papers in which errors for IC50, Ki, or Kd values were higher than 10-15%, for example: Pous et al. (2023) *Nat. Commun.* 14: 7920; Noshi et al. (2018) *Antiviral Res.* 160: 109-117), or even not reported: Lama et al. (2019) *Nat. Commun.* 10: 2261; Gahbauer et al. (2023) *Nat. Commun.* 14: 8067. This includes papers in the LINE-1 and APE1 fields that have reported errors higher than 10-15%, for example: Dai et al. (2012) *BMC Biochem.* 12: 1-11; Zawahir et al. (2009) *J. Med. Chem.* 52: 20-32, or that have not reported errors: Baldwin et al. (2023) *Nature* doi:10.1038/s41586-023-06947-z; Srinivasan et al. (2012) *Biochemistry* 51: 6246-6259. We believe our methodology of performing at least three independent experiments for each inhibitor is rigorous, and the standard deviations between these replicates are sufficient to support our conclusions. For outlier inhibitors with particularly large variations, as pointed out by the reviewer, we have completed additional experiments (replicates) to increase our confidence in these values. These changes have all resulted in decreases in standard deviations.

10. Table 1: How were the concentrations of compound selected for the RetroT efficiencies? The values appear to be a little random and not related to the IC50 values in some cases. For example, AD7 has an IC50 value of 0.875 mM but 20 uM reduced RetroT efficiency to 42%? It's very difficult to argue that with an IC50 value of 0.875 mM, treatment with 20 uM AD7 is on target. The same could be said for AD12. This is very unlikely to be on target.

The concentrations of inhibitors for the retrotransposition assay (Fig. 4, Table 1) were selected based on solubility under cell culture conditions and cytotoxicity measurements by the PrestoBlue Viability Assay (Fig. 4). We agree that off-target effects are a possibility for any of the compounds, and that future work will have to be done to address these issues. Accordingly, we have added a discussion of these potential effects in the Discussion section (lines 341-348). As for the retrotransposition assay, the retrotransposition process is complex and affected by

many host factors, all of which could be potential drug targets. However, we note that we performed three distinct *in vivo* assays: L1 retrotransposition, DNA damage, and senescence-associated inflammation. While the magnitude of the *in vivo* effects varied between the different assays, in our L1-induced DNA damage assays, most of our compounds had measurable and reproducible activity. These experiments were performed by two distinct methods with very different endpoints: γ -H2AX foci and comet assay. We also note that our compounds did not appreciably bind DNA directly at the concentrations tested, while on-target inhibition of the EN domain was shown by multiple *in vitro* assays (Fig. 2, Fig 3, Fig. S3). Hence, while we acknowledge that off-target effects are an important issue, we believe that, given the multitude of *in vivo* assays we used, it is unlikely that all of these effects were due to off-target interactions. For AD7, pointed out by the reviewer, which gave very good inhibition of retrotransposition but relatively weak inhibition of L1-induced DNA damage, an off-target interaction affecting retrotransposition is a reasonable possibility. We note that we also had compounds, such as AD16, that showed good inhibition both *in vitro* and *in vivo*. Finally, we had compounds where *in vitro* inhibition exceeded *in vivo* activity; since none of our compounds were optimized pharmacologically, this can be explained by poor cell penetration or half-life (lines 335-336). Our goal was to characterize our compounds as comprehensively as possible to facilitate future work.

We decided to pursue AD12 in some detail despite its relatively weak inhibition and binding *in vitro* due to its status as a second-generation sulfonylurea drug used for treatment of type 2 diabetes. This suggests that other drugs with a similar mechanism of action may be useful to repurpose for L1 retrotransposition inhibition. The fact that AD12 has shown consistent reduction of L1 activity in multiple cell-based assays (L1 retrotransposition, DNA damage, and senescence-associated inflammation) suggests that this inhibition is the result of interference with the ORF2 protein at least to some extent, rather than purely an off-target effect that is able to modulate these distinct pathways. For example, it is possible that weaker compounds such as AD12 bind the full-length ORF2 protein with higher affinity than the EN domain alone and/or interfere with the dynamics of the full-length ORF2 protein, which we would not detect with *in vitro* experiment testing the EN domain alone. We have added these considerations to the Discussion section (lines 348-351, 376-380).

11. Discussion. It is not possible for good cellular permeability to account for differences in target engagement seen for the compounds as suggested in the following statement. "On the other hand, several inhibitors showed relatively low efficacy *in vitro* but high activity in the retrotransposition assay: AD7, AD9, AD12, AD32, AD36, and AD43. One possibility is that these compounds have higher cell permeability, allowing them to achieve higher intracellular concentrations." How exactly would this happen?

We apologize for our poor wording of this idea. We meant to suggest that the higher cell permeability of some compound such as AD12, a characterized drug, may result in higher concentrations in the cell relative to lower solubility compounds with better IC50s, such as AD11. We do not mean to suggest active transport of the compound into the cell to increase the concentration relative to the concentration outside the cell. We have adjusted the text accordingly (lines 345-351).

12. Table S1. Include CC1/2 values for each data set. Rsym and Rmerge are not the same. Which value is presented in the Table? Also, the Rsym or Rmerge for the highest resolution shell of 6.15 and I/signal of 0.6 for the Mn2+ structure suggests that the data are not usable to the resolution reported. The data should be cut back to lower resolution.

We have added the CC1/2 values and adjusted the resolution cutoff from 1.92Å to 2.23Å; please see our discussion of these points in response to Reviewer 2 (point 4 above). The values reported in Table S1 are Rmerge.

13. Fig. S1, panel A. Showing a surface rendering with no residues labeled or highlighted in different colors is not particularly informative. Either color code residues of interest in the surface rendering or use a semi-transparent surface rendering and show residues of interest highlighted in color as stick renderings.

We have updated Fig. S1 to better show the coordinating residues while still showing where the ligands fall within the DNA binding site used for docking in the surface view. In this context, also see our response to Reviewer 2, point 1.

14. Coordinating ligands for the Mn²⁺ ion should be indicated through dashed lines and bond distances provided in Fig. S1 or in the text.

We have indicated the coordinating residues and waters in Fig. S1 with distance labels included within the figure.

15. What interactions are involved in the binding of tranexamic acid to EN? No hydrogen-bonding interactions are indicated in Figure S1. Fo-Fc density prior to fitting of the fragment should be shown to support the fitting of the fragment.

The predominant interactions with the ligand are with the residue R155 and two waters. We have updated the figure to show these interactions and associated distances. We have also calculated the Fo-Fc difference maps for both structures by omitting the ligands and coordinating waters before calculating the map, which highlights the difference density around the Mn²⁺ and tranexamic acid.

16. The spectral shift binding assay is not standardly used to validate direct binding of small molecules to proteins.

The spectral shift assay is a relatively new method (Langer et al. (2022) *Assay Drug Dev. Technol.* 20: 83-94) that uses similar principles to the related microscale thermophoresis method, but at isothermal conditions. In this method, ligand binding causes a shift in the emission spectrum due to nearby ligand binding changing the chemical environment of the fluorophore conjugated to the receptor protein, and/or larger conformational changes in the protein. This shift is then quantified by calculating the fluorescence intensity ratios of two wavelengths for each inhibitor concentration, which describes the relative amount of bound and unbound protein. These details are included in the Results section (lines 241-249). We have provided the reference in which this was first described in 2022 in the text in the Results (line 242) and Methods (line 575) sections. Our understanding of this method, and the related microscale thermophoresis method, is that they are suitable for the applications in this study.

17. Figures. The figure caption for Figure S2 does not include sufficient information. The text indicates that these compounds are APE1 inhibitors reported in the literature, and yet the compounds are renamed here as AD1-AD15? Then a subset of these compounds are shown docked in EN. Full inhibition of AD3 at 1 mM does not seem like a meaningful result. Many compounds would inhibit at this concentration.

We apologize for the confusion. We have removed the first reference to the docking Figure in the main text that implied that all docked compounds shown were APE1 inhibitors. The docking shown is for all EN inhibitors with *in vitro* efficacy, with the exception of AD1. We have also updated this figure with docking positions for all inhibitors described in Fig. S4 (see above response, point 5). For Fig. S3, we have updated the Figure Legend to clarify that these are EN inhibitors initially screened with the plasmid nicking assay prior to development of the fluorescent oligonucleotide assay. With regard to inhibition of EN activity at 1 mM, we have acknowledged that this represents very weak inhibition in the beginning of the Results section (lines 176-177). At the beginning of this project, in the absence of any existing EN inhibitors, we chose to use this as a starting point to develop improved inhibitors, similar to the use of weak binding fragments for lead optimization. We used the plasmid nicking assay as a qualitative method for screening inhibitors prior to development of the fluorescent oligonucleotide assay, which showed effects of AD2 at lower concentrations than 1 mM. In addition, over the course of this study we have tested many candidate compounds that did not inhibit at 1 mM in the plasmid nicking or fluorescent oligonucleotide assays.

REVIEWER COMMENTS

Reviewer #1 (Remarks to the Author):

My concerns have been addressed - I recommend to accept.

Reviewer #2 (Remarks to the Author):

The revised manuscript is enormously improved. There are only two minor issues that need to be addressed. First, related to my question 7, the value of the σ (sigma) Cutoff of the Fo-Fc omit map in Figure S8 needs to be provided, which is essential for judgment. Second, related to my question 5, the standard deviation of IC50 and Kd must also be supplied in Fig.2 and 3, even though they are one independent assay (experiment) comprised of 3 technical replicates.

Reviewer #3 (Remarks to the Author):

While enthusiasm for the target L1 EN and identification of small molecule inhibitors remains, there are serious problems with some of the conclusions regarding useful small molecule inhibitors. Many of the concerns raised in the initial reviews were addressed including reassessment and refinement of the Mn²⁺ crystal structure, which was initially reported to higher resolution than justified by the data processing statistics; however, the major concerns regarding assessment of small molecule inhibitors were not appropriately addressed.

There are many inconsistencies in the inhibitor characterization data and virtually no discussion of these inconsistencies. Of the inhibitors highlighted as being useful (AD3, AD9, AD12, AD14, AD16, AD29, and AD43), AD9, AD12, and AD43 have IC50 values for L1 EN inhibition ranging from 472-703 μ M but inhibit retrotransposition at much lower concentrations, 25 or 50 μ M. These data are not consistent with inhibition of L1 EN in cells. Additional problems with data include an IC50 of 472 for AD9 but Kd of 5.9 μ M. It is not clear why these very weak inhibitors were used in cell-based assays. Why would you test inhibitors with IC50 values of \sim 500 μ M or greater for inhibition of retrotransposition? And then argue that they target L1 EN in a cell and inhibit retrotransposition. It is not possible to draw this conclusion based on the data. Independent of cell permeability, you would not be able to inhibit the L1 EN at a concentration of 20 μ M if it requires 500 μ M in vitro. This is a major weakness of the revised manuscript.

In this type of study, it is essential to have specific criteria for advancing small molecules. Typically, this would include an IC50 value for the in vitro assay as this was a target-based approach. Small molecules with IC50 values greater than 20 μ M would typically be triaged given that there were a number of compounds with lower IC50 values that look more promising.

Response to Review

In the text below, the comments of the reviews appear in black font, and our response to each point is shown in blue font. We provide line numbers for all the major changes and/or additions we made to the revised manuscript file. In the manuscript file as well as the supplement file substantive changes are highlighted in red font.

Fig.1, panel A was redrawn (Reviewer 3) and Fig. 2 and Fig. 3 have been recalculated (Reviewer 2).

Reviewer #1

My concerns have been addressed - I recommend to accept.

We thank the reviewer for his/her endorsement.

Reviewer #2

The revised manuscript is enormously improved. There are only two minor issues that need to be addressed. First, related to my question 7, the value of the σ (sigma) Cutoff of the Fo-Fc omit map in Figure S8 needs to be provided, which is essential for judgment. Second, related to my question 5, the standard deviation of IC50 and Kd must also be supplied in Fig.2 and 3, even though they are one independent assay (experiment) comprised of 3 technical replicates.

We appreciate the requests of the reviewer and we are happy to make all the requested changes.

The value of the cutoff for the omit map had been provided in the initial submission, but it was labeled as "rmsd". We have changed this label to sigma (σ) in the Figure Legend, so it now reads as "contoured at 3σ "; see Extended Data file, Fig. S1.

Both Fig. 2 and 3 have been reworked as suggested. Each panel shows one representative independent experiment run in triplicate (3 technical replicates). Each replicate had a curve fitted, and all 3 curves (and all data points) are shown in the one panel. The mean of the individual IC50/Kd values from the 3 curves and the s.d. are shown in the panel. This resulted in very small changes in the actual values relative to our previous calculations (one curve fitted to all data points from the 3 technical replicates). For consistency we have recalculated the IC50 and Kd values this way for all independent experiments and updated Table 1 and Table S3 accordingly. Details of these calculations have been added to the Figure Legends (lines 927-930, 937-940) and Methods section (lines 555-557, 574-578).

Reviewer #3

While enthusiasm for the target L1 EN and identification of small molecule inhibitors remains, there are serious problems with some of the conclusions regarding useful small molecule inhibitors. Many of the concerns raised in the initial reviews were addressed including reassessment and refinement of the Mn²⁺ crystal structure, which was initially reported to higher resolution than justified by the data processing statistics; however, the major concerns regarding assessment of small molecule inhibitors were not appropriately addressed.

There are many inconsistencies in the inhibitor characterization data and virtually no discussion of these inconsistencies. Of the inhibitors highlighted as being useful (AD3, AD9, AD12, AD14, AD16, AD29, and AD43), AD9, AD12, and AD43 have IC₅₀ values for L1 EN inhibition ranging from 472-703 μ M but inhibit retrotransposition at much lower concentrations, 25 or 50 μ M. These data are not consistent with inhibition of L1 EN in cells. Additional problems with data include an IC₅₀ of 472 for AD9 but K_d of 5.9 μ M. It is not clear why these very weak inhibitors were used in cell-based assays. Why would you test inhibitors with IC₅₀ values of \sim 500 μ M or greater for inhibition of retrotransposition? And then argue that they target L1 EN in a cell and inhibit retrotransposition. It is not possible to draw this conclusion based on the data. Independent of cell permeability, you would not be able to inhibit the L1 EN at a concentration of 20 μ M if it requires 500 μ M *in vitro*. This is a major weakness of the revised manuscript.

In this type of study, it is essential to have specific criteria for advancing small molecules. Typically, this would include an IC₅₀ value for the *in vitro* assay as this was a target-based approach. Small molecules with IC₅₀ values greater than 20 μ M would typically be triaged given that there were a number of compounds with lower IC₅₀ values that look more promising.

We acknowledge the point of view of the reviewer, and on the advice given we have greatly revised our discussion of the inhibitor characterization data. We sincerely believe the changes we have made accommodate the remaining objections.

First, we would like to address the issue of why compounds with high IC₅₀ values were “advanced” for further study. The compounds were named in numerical order as they emerged from the *in silico* design pipeline. In a historical context, our initial biochemical work was done with the plasmid-nicking assay, which we considered to be semi-quantitative. It took us some time to develop and troubleshoot the more accurate oligo-fluorescence assay. However, the *in vivo* retrotransposition assay has been well established even before we started our study, and is also quite easy and rapid to perform. Thus, quite simply, many of the compounds were tested *in vivo* before we had a firm grip on the IC₅₀ values. We have revised the schematic representation of our work flow accordingly, see Fig. 1, panel A.

Second, we have provided a possible explanation for the AD9 discrepancy (IC₅₀ vs. K_d value) pointed out by the reviewer (lines 255-257). That whole section in the Results (Biophysical characterization of EN inhibitors) has been reworked and expanded (lines 250-257).

Third, at the advice of the reviewer, we have removed AD9, AD12, and AD43 from the list of “useful” molecules (line 351). We have included language in the Discussion, as requested by the reviewer, that these IC₅₀ “data are not consistent with inhibition of L1 EN in cells” (lines 347-348). We have removed the paragraph in which we “argue that they target EN in a cell”. Finally, we have provided an expanded summary of our inhibitor data, and further interpretation of the findings, along the lines suggested by the reviewer (lines 338-356).

We thank the reviewer for his/her rigorous review of our work, which we believe has greatly improved the quality of the paper.

REVIEWERS' COMMENTS

Reviewer #3 (Remarks to the Author):

The authors have now addressed the concerns noted in the second review of the manuscript.